# Tree-Based Premise Selection for Lean4

**Zichen Wang**
School of Mathematical Sciences
Peking University, China
princhernwang@gmail.com

**Anjie Dong**
School of Data Science
The Chinese University of Hong Kong, Shenzhen, China
anjiedong@link.cuhk.edu.cn

**Zaiwen Wen**
Beijing International Center for Mathematical Research
Peking University, China
wenzw@pku.edu.cn

## Abstract

Premise selection is a critical bottleneck in interactive theorem proving, particularly with large libraries. Existing methods, primarily relying on semantic embeddings, often fail to effectively leverage the rich structural information inherent in mathematical expressions. This paper proposes a novel framework for premise selection based on the structure of expression trees. The framework enhances premise selection ability by explicitly utilizing the structural information of Lean expressions and by means of the simplified tree representation obtained via common subexpression elimination. Our method employs a multi-stage filtering pipeline, incorporating structure-aware similarity measures including the Weisfeiler-Lehman kernel, tree edit distance, `Const` node Jaccard similarity, and collapse-match similarity. An adaptive fusion strategy combines these metrics for refined ranking. To handle large-scale data efficiently, we incorporate cluster-based search space optimization and structural compatibility constraints. Comprehensive evaluation on a large theorem library extracted from Mathlib4 demonstrates that our method significantly outperforms existing premise retrieval tools across various metrics. Experimental analysis, including ablation studies and parameter sensitivity analysis, validates the contribution of individual components and highlights the efficacy of our structure-aware approach and multi-metric fusion.

## 1 Introduction

Automated Theorem Proving (ATP) is a core research direction in the fields of Artificial Intelligence and Formal Methods [1, 2]. When constructing formal proofs, especially when faced with vast and continuously evolving libraries of mathematical theorems, a key challenge is efficiently identifying relevant theorems, lemmas, and definitions from a massive pool of candidates. This task is known as Premise Selection [2, 3]. Efficient and accurate premise selection can significantly narrow down the effective search space for subsequent proof generation, and it represents a critical bottleneck in achieving practical automated and interactive theorem proving systems. In recent years, with the development of Large Language Models (LLMs), Retrieval-Augmented Generation (RAG) architectures have garnered attention in the theorem proving domain [4]. In this architecture, premise selection serves as the preceding retrieval step, and its performance directly impacts the success rate of subsequent proof generation.

A specific type of formal system is called a proof assistant, also known as an interactive theorem prover. These software tools are designed to assist human users in constructing and verifying mathematical proofs or verified software in a strictly formalized manner. Popular proof assistants

39th Conference on Neural Information Processing Systems (NeurIPS 2025).

currently include, Coq [5], Isabelle [6], HOL Light [7], Metamath [8], and the increasingly prominent Lean [9, 10] in recent years. Notably, Lean, with its powerful type system and active community, has fostered comprehensive libraries such as Mathlib [11] and Optlib [12–14], containing a large amount of formalized mathematical knowledge. Efficient premise selection within these proof assistants possessing large theorem libraries is crucial for users performing formalization work. Currently, in the Lean formal proof assistant, mainstream premise selection tools, such as Lean State Search [15], Loogle [16], Moogle [17], Lean Explore [18] and Lean Search [19], typically employ methods based on semantic embeddings of target states, theorem statements, or mathematical expressions. They often fail to fully utilize the rich structural information inherent in mathematical expressions.

Given the inherent structured nature of mathematical formulas, machine learning research in the field of theorem proving began exploring how to leverage the underlying structure of formal statements for representation learning quite early on. This direction typically involves parsing formulas into structured forms such as trees or graphs and applying corresponding machine learning models[20–23]. Early attempts combined formula structure with Convolutional Neural Networks (CNNs) or Recurrent Neural Networks (RNNs) [24, 25] and evaluated their capabilities on tasks such as proof strategy development, premise selection [26], and capturing and utilizing the structure of logical expressions [27] on datasets like HOLStep [24]. As research deepened, neural network models specifically designed for structured data, particularly tree-structured neural networks and Graph Neural Networks (GNNs), were more widely used to encode the structural information of formal statements [26, 28–36]. These structured methods aim to capture logical invariances in formulas, improve graph representations [31], and have been applied to tasks such as higher-order logic proof search [29]. Furthermore, explorations utilizing structural information also include graph contrastive learning [37–40] and methods built upon theorem dependency graphs [41, 42]. Collectively, these works demonstrate that explicitly modeling and utilizing the structural information of formal mathematical statements is crucial for improving the performance of machine learning in theorem-proving related tasks. Although existing works have made important explorations in utilizing structural information [43, 44], designing a structured method for proof assistants like Lean, which have complex type systems and large libraries, that can both accurately capture expression structure and efficiently perform premise selection, remains a problem requiring in-depth research.

To address the above challenges, we propose a novel tree-based premise selection framework for Lean4. The framework explicitly leverages the structural information of Lean expressions, aiming for efficient and accurate retrieval of relevant premises from large theorem libraries. We first apply Common Subexpression Elimination (CSE) to standardize expressions into trees, and then use a two-stage pipeline: a computationally efficient, training-free Weisfeiler–Lehman (WL) kernel for rapid coarse screening, followed by Tree Edit Distance (TED) with a principled fusion of structural/semantic similarities for fine ranking. This unified design principle—*structure first, semantics as complement*—turns multi-metric fusion into a consistent decision rule rather than ad-hoc stacking. To keep costs scalable, we separate offline fingerprinting/clustering from online retrieval and employ bounded/pruned TED together with a query-adaptive candidate budget. While instantiated in Lean4, the backend is *language-lean*: we envision the design of a lightweight adapter to facilitate portability to other proof assistants that expose (or allow reconstruction of) typed, binder-aware expression trees. We evaluate under a unified protocol and release code/configuration for reproducibility. Our main contributions are:

- **Structured theorem library and benchmark construction.** We build a tree-structured representation of Mathlib4 and design two formal test sets for premise selection in Lean4, enabling systematic and reproducible comparisons.
- **Adaptive multi-metric similarity fusion.** We introduce a principled fusion of WL-cosine, TED, constant-level Jaccard, and Collapse-Match alignment under a structure-compatibility constraint, capturing both global shape and local edit consistency.
- **Scalable tree-based retrieval framework.** We integrate CSE normalization, WL coarse filtering, and cluster-based optimization with *bounded/pruned* TED and a query-adaptive candidate budget, achieving superior performance across multiple retrieval metrics.

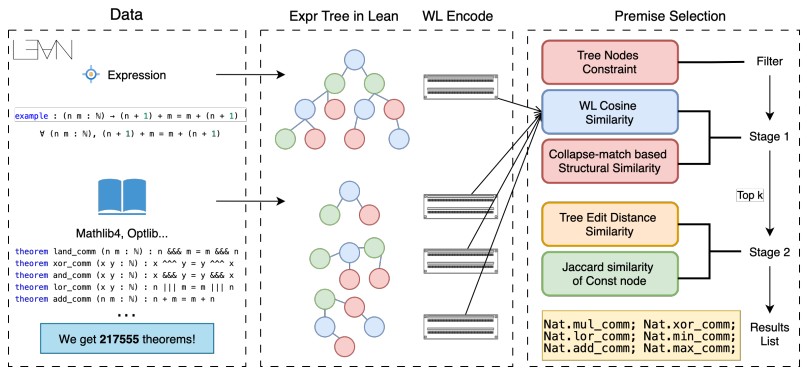

Figure 1: Overview flowchart of our proposed tree-based premise selection method

## 2 Preliminaries

### 2.1 Premise Selection for Lean4

Premise selection is a critical step in Automated Theorem Proving (ATP) and Interactive Theorem Proving (ITP), aimed at retrieving the most relevant theorems and lemmas from a vast theorem library that are related to the current proof goal [2, 3]. Efficient and accurate identification of relevant premises is crucial for managing large theorem libraries and reducing the proof search space. Lean4 is an advanced interactive theorem proving system known for its powerful functional programming capabilities and its continuously growing large mathematical library [9, 10]. In Lean4, users decompose proof goals by applying tactics (`apply`, `rw`, etc.) and existing theorems/lemmas. As the Lean mathematical library continues to expand, the role of premise selection in enhancing the efficiency and scalability of the proof process becomes increasingly prominent. The complex tree structure and rich type system of Lean expressions pose challenges for traditional text-based or simple feature-based similarity matching methods, simultaneously highlighting the need for structure-aware and efficient retrieval methods.

### 2.2 Expr Tree in Lean4

In Lean4, the `Expr` data structure is the core representation for formal mathematical formulas and proofs. It is a recursive structure that forms what is known as an expression tree (`Expr` Tree). Lean4 uses this structure to represent and process various linguistic constructs, including bound variables (`bvar`), free variables (`fvar`), constants (`Const`), function applications(`App`), lambda abstractions (`lam`), and quantified expressions, enabling efficient parsing, manipulation, and type checking of formulas. The `Expr` Tree maps complex mathematical formulas into a hierarchical structure: the root node represents the outermost expression, while each child node represents a component of the formula. This recursive and hierarchical nature allows Lean4 to efficiently process, verify, and reason about formulas recursively. The node types of the expression tree are defined by the various constructors of the `Expr` structure, such as `bvar`, `fvar`, and `lam`, each storing necessary information related to the corresponding expression type.

For instance, the mathematical expression $\mathbb{N} \rightarrow \texttt{Prop} : x \mapsto \forall y \in \mathbb{N}, x = y$, is represented in Lean4 as `fun x:Nat => forall y:Nat, x = y`. This expression is transformed into a tree structure, as shown in Figure 2. The recursive nature of the expression tree is fundamental to Lean4's efficient processing and reasoning about formulas.

Leveraging expression trees allows theorems in Lean4 to be modeled as structured trees. This hierarchical tree structure clearly expresses the internal composition of theorems and provides a basis for measuring the similarity between different theorems. By comparing the similarity of these expression trees, we can effectively quantify the structural and potential semantic relevance between theorems, which is crucial for the automated processing and retrieval of theorems. Additionally, the proposed framework is language-lean, meaning that it is adaptable to other proof assistants, such

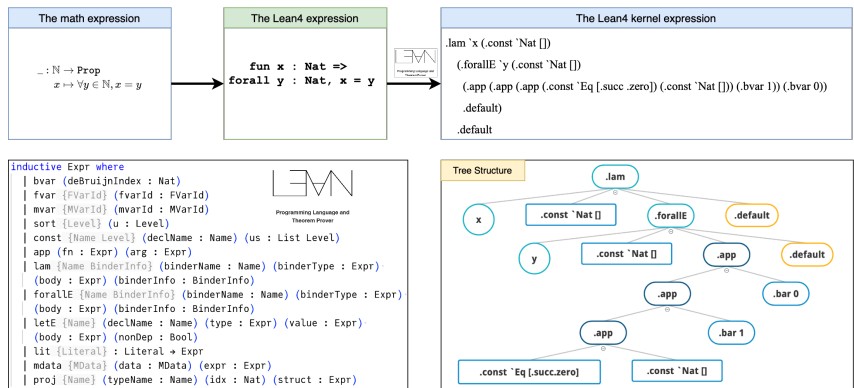

Figure 2: Expression tree representation in Lean4. The diagram shows the translation of a mathematical expression into the corresponding Lean4 expression and its representation as an expression tree. The definition of `Expr` in Lean4, as shown in the bottom left corner, includes constructors like `bvar`, `fvar`, `lam`, and `app`.

as Coq and Isabelle, provided that these systems can export or reconstruct a similar expression tree structure.

# 3 Expression to Tree for premise selection in Lean4

## 3.1 Expressions to Trees with CSE

In the simplification of expression trees, a crucial task is the elimination of common subexpressions. The CSE algorithm helps to reduce redundancy in an expression by ensuring that each subexpression is evaluated only once. The process can be broken down into several steps that systematically identify and replace repeated subexpressions. A simple example is presented in Figure 3.

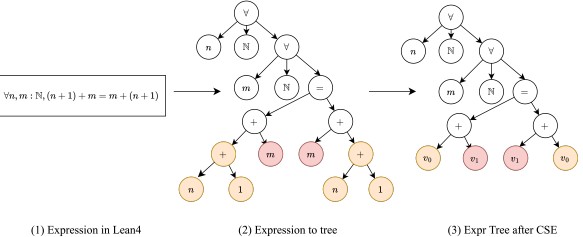

(1) Expression in Lean4     (2) Expression to tree     (3) Expr Tree after CSE

Figure 3: Applying CSE to an expression tree

Next, we provide a rigorous and formal explanation of the CSE algorithm adopted for Lean4's `Expr` structure. This algorithm primarily comprises the following three sequential stages, aimed at standardizing and simplifying the expression tree structure by replacing repeated subexpressions. We use symbolic functions $\mathcal{D}, \mathcal{C}, \mathcal{R}$ and auxiliary data structures to describe these stages and their interactions.

**1. Processing de-Bruijn Indices** ($\mathcal{D}$). This stage, performed by function $\mathcal{D}$, recursively traverses the expression tree to standardize the representation of bound variables (`bvar`), ensuring it is unaffected by local binder names for facilitating subsequent subexpression comparison. The output is the expression tree $e' = \mathcal{D}(e, \varnothing)$ with standardized de-Bruijn indices.

**2. Counting Subexpression Occurrences** ($\mathcal{C}$). The objective of this stage operates on the output $e'$ from the first stage. Function $\mathcal{C}$ traverses the tree, computes the structural hash value for each subexpression, and uses a global dictionary $\mathcal{M}$ to record its frequency of appearance, thereby identifying repeated subexpressions. The output is the dictionary $\mathcal{M}$ containing subexpression occurrence counts.

**3. Replacing Repeated Subexpressions** ($\mathcal{R}$). This stage receives the standardized expression tree $e'$ and the frequency dictionary $\mathcal{M}$. Function $\mathcal{R}$ recursively traverses $e'$, replacing subexpressions whose hash values appear more than once (excluding constants) with newly introduced free variables (`fvar`). Internal mapping $\mathcal{V}$ and set $S$ are used for variable management. The output is the CSE-simplified expression tree.

The entire CSE algorithm $\text{cse}(e)$ can be understood as the original expression $e$ undergoing the sequential processing of the three stages described above: $\text{cse}(e) = \mathcal{R}(\mathcal{D}(e, \varnothing), \mathcal{M}_{\text{final after } \mathcal{C}}, \mathcal{V}_{\text{init}}, S_{\text{init}})$. In this formula, $\mathcal{D}(e, \varnothing)$ is the output of the first stage, $\mathcal{M}_{\text{final after } \mathcal{C}}$ is the frequency dictionary populated after running the second stage $\mathcal{C}$ on $\mathcal{D}(e, \varnothing)$, while $\mathcal{V}_{\text{init}}, S_{\text{init}}$ are the initial empty map and set at the beginning of the third stage. After being processed by CSE, the original expression $e$ is transformed into an expression tree with a more compact and less redundant structure, where common subexpressions are replaced by unified variable references. This standardized representation significantly increases the efficiency of processing and analyzing expressions in Lean4 theorem proving tasks, providing an optimized input for subsequent structural similarity calculations and premise selection procedures.

## 3.2 The WL Kernel Method for Encoding

The WL kernel is a graph kernel method used to measure structural similarity between graphs[45]. It iteratively aggregates node neighborhood information to generate graph representations, effectively capturing graph topological features. The WL kernel has been proven to be highly capable of distinguishing non-isomorphic graphs. Given an expression tree $T = (V, E)$, where $V$ is the set of nodes and $E$ is the set of edges, the WL kernel method encodes the tree through the following iterative process.

1. **Initialization**: Assign each node $v \in V$ an initial label $l_v^{(0)}$, typically based on its node type (e.g., variable, constant, or function symbol).
2. **Iteration**: For each iteration $i = 1, 2, \ldots, N$ ($N$ is the predefined number of iterations):
   - For each node $v$, collect the multiset of current labels from its neighbor set $N(v)$, $\{l_u^{(i-1)} \mid u \in N(v)\}$.
   - Combine the node's current label $l_v^{(i-1)}$ with the multiset of its neighbors' labels to form a string, for example, $l_v^{(i-1)}$"-"sorted_multiset($l_u^{(i-1)}$ for $u \in N(v)$).
   - Use a deterministic hash or compression function to map this combined string to a new, unique label $l_v^{(i)}$. Update the node's label to $l_v^{(i)}$.

   In each iteration, record the set of new labels for all nodes.
3. **Feature Vector**: After $N$ iterations, compute a feature vector for tree $T$ by counting the frequency of all labels that appeared across all iterations.

The WL kernel is closely related to message-passing neural network (MPNN) mechanisms [23], and its 1st-order test (1-WL) has been proven to be highly capable of distinguishing graph structures [46]. This powerful structural discriminative capability makes the WL kernel particularly suitable for encoding Lean4 expression trees, which have complex and unique structures, effectively capturing local structural patterns and generating discriminative vector representations. The WL kernel value is defined as the inner product of two graphs' feature vectors $k_{\text{WL}}(T_1, T_2) = \langle \phi(T_1), \phi(T_2) \rangle$ and similarity based on feature vectors (such as cosine similarity) can quantify structural similarity between trees. In the premise selection method proposed in this paper, we utilize the WL kernel method to encode CSE-simplified expression trees. These feature vectors are used in the initial screening phase to compute and rank the structural similarity between the query and library theorems, selecting a Top-$k$ candidate set, thus efficiently leveraging structural information in the preliminary stage.

## 3.3 Tree Edit Distance Similarity

TED is a common measure for structural similarity between ordered labeled trees, defined as the minimum total weighted cost required to transform one tree into another using insertion, deletion, and substitution operations [47]. Computing ordered tree TED is a classic problem, often solved by dynamic programming algorithms. The zss algorithm [48] is a classic method with a time complexity of $O(|T_1| \times |T_2| \times \min(\text{depth}(T_1), \text{leaves}(T_1)) \times \min(\text{depth}(T_2), \text{leaves}(T_2)))$ and space complexity $O(|T_1| \times |T_2|)$, theoretically efficient and supporting parallelization.

In our application to Lean4 expression trees, editing operation costs $C_R$ (substitution), $C_D$ (deletion), $C_I$ (insertion) are finely tuned based on Lean characteristics. Specifically, costs are set as follows. Define a set of node label prefixes with smaller change costs $\mathcal{P} = \{\texttt{BVar}, \texttt{FVar}, \texttt{MVar}, \texttt{Sort}, \texttt{Const}\}$. For node substitution $C_R$, the cost is 0 if the two nodes are identical or if the label of either node starts

with any string in $\mathcal{P}$; otherwise, the cost is the preset value $C_R$. For node insertion ($C_I$) or deletion ($C_D$), if the involved node's label starts with any string in $\mathcal{P}$, the cost is $\sigma C_I$ or $\sigma C_D$, respectively, where $\sigma \in [0,1]$ is a scaling parameter for simple node operations; otherwise, the cost is the preset value ($C_I$ or $C_D$). This tiered cost structure reflects the differing impacts of operations on expression structure and logical core. Operations on simple nodes representing variables, types, or constants (low cost) typically have less impact on overall mathematical meaning compared to modifications of complex expression structures (high cost). By defining cost functions adapted to the semantics and structural characteristics of Lean4 expressions, TED can more precisely capture subtle and logically relevant structural differences between trees. Based on the TED $d(T_1, T_2)$, the normalized similarity tree edit distance similarity (TEDS) $s(T_1, T_2)$ is defined as:

$$s(T_1, T_2) = 1 - \frac{d(T_1, T_2)}{C \cdot \max(|T_1|, |T_2|)}, \tag{1}$$

where $|T|$ is the node count, and $C = \max(C_R, C_D, C_I)$ is the normalization factor ensuring $s \in [0, 1]$. In the premise selection method proposed in this paper, TEDS serves as the fine-grained metric in the second stage, applied to CSE-simplified expression trees. After WL kernel method preliminarily screens Top-$k$ candidates, TEDS scores are computed against the target tree for re-ranking. TED's fine-grained comparison aids more accurate relevance assessment, optimizing final premise selection.

## 4 Tree-Based Premise Selection

### 4.1 Core Tree-Based Selection with Multi-Stage Filtering

The core premise selection mechanism proposed in this paper employs a two-stage filtering pipeline designed for efficient retrieval of the most relevant theorems from a large theorem library, balancing computational efficiency and retrieval accuracy. The inputs are the target expression tree $T_{\text{query}}$ and the premise theorem database $S_{\text{CSE}}$, where each theorem is preprocessed via CSE and represented as an expression tree.

**Stage 1: Coarse Filtering via WL Kernel**
The objective of this stage is to rapidly reduce the search space. We first encode the target expression tree $T_{\text{query}}$ and all theorem trees $T_i \in S_{\text{CSE}}$ using the Weisfeiler-Lehman (WL) kernel method (with $h$ iterations, see Appendix B), generating feature vectors $\phi_{\text{WL}}(\cdot) \in \mathbb{N}^d$ that reflect the tree's structural features. Subsequently, we compute the cosine similarity between the WL feature vector of the target expression tree $\phi_{\text{WL}}(T_{\text{query}})$ and the feature vector $\phi_{\text{WL}}(T_i)$ of each theorem tree $T_i$ in the database: $\text{sim}_{\cos}(T_{\text{query}}, T_i) = \frac{\phi_{\text{WL}}(T_{\text{query}}) \cdot \phi_{\text{WL}}(T_i)}{\|\phi_{\text{WL}}(T_{\text{query}})\| \|\phi_{\text{WL}}(T_i)\|}$. Based on the cosine similarity scores, we rank the theorems in the database and select the top $k$ theorems as the coarse candidate set $C_{\text{coarse}} = \{T_1, \ldots, T_k\}$. The parameter $k$ is a tunable threshold balancing recall and the computational overhead of subsequent refined filtering. To adapt to target expressions of varying complexity, the value of $k$ can be adjusted adaptively based on the number of nodes $|T_{\text{query}}|$ in $T_{\text{query}}$: $k = \min\left(k_{\max}, \beta \cdot |T_{\text{query}}|\right)$, where $k_{\max}$ is the upper bound for $k$, and $\beta$ is a scaling factor used to determine the initial number of candidates based on the target tree size.

**Stage 2: Refined Filtering via TEDS**
This stage aims to perform a more precise structural similarity assessment on the coarse candidate set $C_{\text{coarse}}$. For each theorem tree $T_j \in C_{\text{coarse}}$ in the candidate set, we compute the normalized TEDS score between it and the target expression tree $T_{\text{query}}$, as shown in Eq. 1: $\text{sim}_{\text{TEDS}}(T_{\text{query}}, T_j) = 1 - \frac{d(T_{\text{query}}, T_j)}{C \cdot \max(|T_{\text{query}}|, |T_j|)}$. The final similarity score for the core method combines the results from both stages, using a weighted sum to calculate the hybrid similarity: $\text{sim}_{\text{hybrid}}(T_{\text{query}}, T_j) = \alpha \cdot \text{sim}_{\cos}(T_{\text{query}}, T_j) + (1 - \alpha) \cdot \text{sim}_{\text{TEDS}}(T_{\text{query}}, T_j)$, where $\alpha \in [0, 1]$ is a weight parameter controlling the relative importance of WL cosine similarity and TEDS in the final score. The coarse candidate set $C_{\text{coarse}}$ is re-ranked based on the hybrid similarity $\text{sim}_{\text{hybrid}}$, and the top $m$ theorems are selected as the final premise selection result for the core method, to be used for subsequent tactic application or proof search.

## 4.2 Enhanced Selection via Clustering and Multi-Strategy Expansion

Building upon the core two-stage filtering pipeline proposed in Section 4.1, this subsection introduces several enhanced mechanisms to further improve the efficiency, accuracy, and flexibility of premise selection. These enhanced mechanisms include cluster-based search space optimization, structural compatibility constraint filtering, additional structural simplification, collapse-match similarity, Jaccard similarity of `Const` nodes and an adaptive similarity metric fusion strategy, collectively forming a more powerful premise selection framework.

**Cluster-Based Search Space Optimization.** To accelerate the initial screening process for a large premise theorem database $S_{\text{CSE}}$, we employ cluster-based search space optimization techniques. The core idea is to group theorems based on their structural features (encoded via the WL kernel method and converted into vector representations, typically after dimensionality reduction), thereby restricting the search scope during online retrieval to within the cluster(s) most relevant to the query expression $T_{\text{query}}$, significantly reducing computational overhead. This mechanism comprises an offline phase, where library theorem vectors are clustered into $K$ groups and centroids are computed; and an online phase, where the query expression vector's similarity to each cluster centroid is computed to select relevant clusters for restricting the Stage 1 search. For dynamically updated theorem libraries, maintenance mechanisms (such as incremental updates or periodic re-clustering) should be considered to ensure the effectiveness of the clustering structure. Appendix C show the detail.

**Structural Compatibility Constraint Filtering.** Before performing detailed structural similarity calculations, we introduce structural compatibility constraints based on the node count $|T|$ to preliminarily filter candidate theorems, aiming to exclude theorems differing significantly in size from the target expression tree $T_{\text{query}}$, thereby improving efficiency and reducing false positives. Only theorems satisfying the following set of size constraints $\mathcal{C}_{\text{valid}}$ proceed to the next stage of detailed comparison: $\mathcal{C}_{\text{valid}} = \underbrace{\left\{ T \mid c_1|T_{\text{query}}| \leq |T| \leq c_2|T_{\text{query}}| \right\}}_{\text{Relative Size Constraint}} \cap \underbrace{\left\{ T \mid \big||T| - |T_{\text{query}}|\big| \leq \Delta \right\}}_{\text{Absolute Size Constraint}}$, where $c_1 < 1 < c_2$ are constants defining relative node count boundaries, and $\Delta$ specifies the maximum allowed absolute difference in node count. Only theorems belonging to $\mathcal{C}_{\text{valid}}$ proceed to the detailed comparison stage.

**Additional Structural Simplification.** Given the sensitivity of TED to structural details, we apply an additional structural simplification step to expression trees before computing the TEDS. This step aims to remove redundant binding structures that are structurally similar but differ little in their logical core, particularly bindings involving simple types such as `BVar`, `FVar`, `MVar`, `Sort`, `Const`. Specifically, this process recursively traverses the CSE-simplified expression tree. For a `ForallE` node, if its `binderType` belongs to these simple types, the node is replaced with its `body`, and this simplification iterates until the tree structure no longer changes. This additional simplification helps to reduce TEDS's sensitivity to these secondary structural changes, allowing the computed TEDS to better reflect the similarity of the expression's core logical structure.

**Collapse-Match (CM) Similarity.** $\text{sim}_{\text{CM}}(T_a, T_b)$ is a structural similarity metric designed to measure the degree to which the structure of the source tree $T_a$, after simplification according to specific collapse rules, aligns with the target structure $T_b$. Its computation method involves calculating a raw match score for $T_a$ aligning with $T_b$ through a recursive process and normalizing this score by dividing by the total number of nodes $|T_b|$ in the target tree $T_b$, yielding a score between 0 and 1. This metric is particularly suitable for evaluating whether a premise structure can match a segment of the proof goal structure by applying specific proof rules or tactics. See Appendix D for the details.

**Jaccard Similarity of `Const` nodes.** $\text{sim}_{\text{Jaccard}}(T_a, T_b)$ measures the semantic overlap between two expression trees $T_a$ and $T_b$, specifically based on the sets of `declNames` of their `Const` nodes. The computation involves collecting all unique `declNames` of `Const` nodes in $T_a$ and $T_b$ to form sets $S_a$ and $S_b$, respectively, and then computing the sizes of their intersection and union. It is defined as: $\text{sim}_{\text{Jaccard}}(T_a, T_b) = \frac{|S_a \cap S_b|}{|S_a \cup S_b|}$. When the union is empty, the similarity is 1 if $S_a, S_b$ are both empty, otherwise 0. The details are provided in Appendix D.

**Adaptive Similarity Metric Fusion.** We generalize the similarity fusion strategy from the core method, employing a more flexible parametric approach that considers multiple similarity metrics. The final method fuses four structural and semantic similarity metrics introduced in the preceding text. The final enhanced similarity score $\text{sim}_{\text{enhanced}}$ is computed via the following weighted sum:

$$\text{sim}_{\text{enhanced}} = \alpha \cdot \text{sim}_{\text{cos}} + \beta \cdot \text{sim}_{\text{TEDS}} + \gamma \cdot \text{sim}_{\text{Jaccard}} + \delta \cdot \text{sim}_{\text{CM}}, \tag{2}$$

where $\alpha, \beta, \gamma, \delta$ are weight parameters satisfying $\alpha + \beta + \gamma + \delta = 1$, dynamically adjusted based on the characteristics of the target expression and candidate theorems to optimize retrieval performance. Through this adaptive fusion, we can dynamically adjust the importance of different similarity metrics, achieving more precise premise selection based on target expression and candidate theorem characteristics. Overall, our method employs a two-stage strategy: Stage 1 performs rapid coarse screening by combining tree node count, collapse-match based structural alignment similarity, and computationally efficient WL cosine similarity; Stage 2 then utilizes this comprehensive adaptive fused similarity for refined ranking of the preliminary results to obtain the final premise recommendations.

## 5 Numerical Experiments

### 5.1 Dataset Preparation

The theorem library is constructed by extracting 217,555 valid mathematical theorems from Lean4's Mathlib4 version v4.18.0-rc1. The structural characteristics and domain distribution of the theorem library are evaluated by statistical analysis, with results visualized and detailed complexity and domain statistics provided in Appendix A. Each theorem tree in the CSE-preprocessed database is encoded using the WL kernel method with iteration count $k$, and the encodings are stored in the database. We construct two test sets for the evaluation of the method. The test set A consists of 100 small-scale problems, categorized into substitution-type, condition-swapping-type, and mixed-type, aimed at assessing the method's ability to recognize structural similarities, adapt to premise reordering, and handle combined transformations. The test set B consists of $m = 6119$ (state, theorem) pairs extracted from the Lean proof environment, where each pair transforms the proof state prior to the application of the theorem into a problem statement. See Appendix A for the test set details.

### 5.2 Theorem/Lemma Retrieval Experiments

We compare our method with existing Lean premise retrieval tools, including Lean Search [19], Moogle [17], Lean Explore [18] and Lean State Search [15], as baseline methods. The experiments are carried out on a local machine (Macbook Pro) equipped with an Apple M3 Pro chip (11 cores). The objective is to compare the retrieval performance of the proposed method with the baseline methods in test sets A and B. Performance is evaluated using common Top-k retrieval metrics: Top-k Recall, Top-k Precision, Top-k F1-score, Top-k nDCG, and Mean Reciprocal Rank. We present in Appendix F examples where query expressions and their corresponding matched theorems are depicted as tree structures. For these experiments, the weights were set to $(\alpha, \beta, \gamma, \delta) = (0.1, 0.3, 0.1, 0.5)$, and the TED costs were $C_I = C_D = 1$ and $C_R = 0.4$, with a low-cost scaling factor of $\sigma = 0.2$. The performance metrics for test sets A and B are summarized in Table 1. Experimental results demonstrate that the proposed tree-based method achieves retrieval performance significantly superior to existing baseline tools on both test sets. This is particularly evident on the large test set B, validating the effectiveness of our structured approach for handling complex mathematical expressions. Table 2 and 3 further presents Recall@k performance on Test Set B, stratified by mathematical domain and complexity (node count: simple < 50, medium 50-200, complex > 200). Analysis indicates that retrieval performance varies significantly across different domains and generally decreases with increasing problem complexity.

Table 1: Performance (%) on test sets A and B

| Test Set | Method | Recall@k | | | Precision@k | | | F1-score@k | | | nDCG@k | | | MRR |
|---|---|---|---|---|---|---|---|---|---|---|---|---|---|---|
| | | @1 | @5 | @10 | @1 | @5 | @10 | @1 | @5 | @10 | @1 | @5 | @10 | |
| A | Lean Search | 1.0 | 3.0 | 5.0 | 1.0 | 0.6 | 0.5 | 1.0 | 1.0 | 0.9 | 1.0 | 2.1 | 2.7 | 2.0 |
| | Moogle | 7.0 | 11.0 | 11.0 | 7.0 | 2.2 | 1.1 | 7.0 | 3.7 | 2.0 | 7.0 | 9.4 | 9.4 | 8.9 |
| | Lean Search (Augment) | 1.0 | 4.0 | 8.0 | 1.0 | 0.8 | 0.8 | 1.0 | 1.3 | 1.5 | 1.0 | 2.5 | 3.7 | 2.7 |
| | Lean Explore | 4.0 | 15.0 | 21.0 | 4.0 | 3.0 | 2.1 | 4.0 | 5.0 | 3.8 | 4.0 | 9.6 | 11.5 | 9.6 |
| | Lean State Search | 45.0 | 83.0 | 85.0 | 45.0 | 16.6 | 8.5 | 45.0 | 27.7 | 15.5 | 45.0 | 65.3 | 66.0 | 60.2 |
| | **Tree-based Search** | **82.0** | **84.0** | **86.0** | **82.0** | **16.8** | **8.6** | **82.0** | **28.0** | **15.6** | **82.0** | **83.1** | **83.7** | **83.3** |
| B | Lean Search | 7.2 | 15.2 | 18.0 | 7.2 | 3.0 | 1.8 | 7.2 | 5.1 | 3.3 | 7.2 | 11.4 | 12.3 | 10.8 |
| | Moogle | 4.8 | 10.9 | 13.6 | 4.8 | 2.2 | 1.4 | 4.8 | 3.6 | 2.5 | 4.8 | 8.0 | 8.8 | 7.5 |
| | Lean Search (Augment) | 6.6 | 14.2 | 16.7 | 6.6 | 2.8 | 1.7 | 6.6 | 4.7 | 3.0 | 6.6 | 10.6 | 11.4 | 10.1 |
| | Lean Explore | 1.0 | 7.7 | 13.0 | 1.0 | 1.5 | 1.3 | 1.0 | 2.6 | 2.4 | 1.0 | 4.3 | 6.0 | 4.4 |
| | Lean State Search | 8.2 | 17.6 | 22.0 | 8.2 | 3.5 | 2.2 | 8.2 | 5.9 | 4.0 | 8.2 | 13.2 | 14.6 | 12.8 |
| | **Tree-based Search** | **25.8** | **28.0** | **28.7** | **25.8** | **5.6** | **2.9** | **25.8** | **9.4** | **5.2** | **25.8** | **27.0** | **27.2** | **26.9** |

Table 2: Domain-Complexity Cross-Performance (%)

| Domain | Complexity | Count | Recall@k | | |
|---|---|---|---|---|---|
| | | | @1 | @5 | @10 |
| Algebra | Simple | 47 | 4.3 | 4.3 | 4.3 |
| | Medium | 346 | 1.7 | 2.3 | 2.6 |
| | Complex | 240 | 0.0 | 0.4 | 0.4 |
| Natural Number | Simple | 75 | 68.0 | 81.3 | 82.7 |
| | Medium | 126 | 38.9 | 42.1 | 44.4 |
| | Complex | 21 | 23.8 | 38.1 | 38.1 |
| Logic | Simple | 22 | 36.4 | 40.9 | 40.9 |
| | Medium | 22 | 18.2 | 22.7 | 22.7 |
| | Complex | 24 | 20.8 | 20.8 | 20.8 |

Table 3: Domain-Complexity Cross-Performance (%)

| Domain | Complexity | Count | Recall@k | | |
|---|---|---|---|---|---|
| | | | @1 | @5 | @10 |
| Order Theory | Simple | 136 | 7.4 | 7.4 | 7.4 |
| | Medium | 351 | 4.8 | 5.4 | 5.4 |
| | Complex | 238 | 1.3 | 2.1 | 2.1 |
| Set Theory | Simple | 34 | 52.9 | 55.9 | 58.8 |
| | Medium | 31 | 35.5 | 38.7 | 38.7 |
| | Complex | 3 | 0.0 | 0.0 | 0.0 |
| Others | Simple | 577 | 17.5 | 21.3 | 21.5 |
| | Medium | 2976 | 41.0 | 43.3 | 44.3 |
| | Complex | 762 | 9.2 | 11.5 | 12.1 |

Table 4: Framework hyperparameters and their roles.

| Parameter | Stage | Description |
|---|---|---|
| $h$ | WL Kernel | Iteration count for neighborhood aggregation |
| $\alpha, \beta, \gamma, \delta$ | Adaptive Fusion | Weight parameters for cosine, TEDS, Jaccard, and Collapse-Match similarities |
| $\sigma$ | TEDS Operation Costs | Low-cost scaling factor for simple node operations |

## 5.3 Ablation Study

**Hyperparameters and Their Impact.** We introduce several key hyperparameters that govern the premise selection process. Table 4 summarizes these hyperparameters and their roles.

- WL Kernel Iteration Count ($h$): The number of iterations in the Weisfeiler-Leman kernel determines the depth of neighborhood aggregation. Larger $h$ captures more global structural features but increases computation and may cause oversmoothing. We select $h$ to balance efficiency with structural expressiveness.
- Adaptive Fusion Weight Parameters ($\alpha, \beta, \gamma, \delta$): These parameters control the relative contributions of cosine similarity, tree edit distance similarity, Jaccard similarity, and Collapse-Match similarity. They satisfy $\alpha + \beta + \gamma + \delta = 1$ and are dynamically adjusted to optimize retrieval performance.
- Low-Cost Scaling Factor ($\sigma$): In TEDS computation, operations on simple nodes (e.g., BVar, FVar, MVar, Sort, Const) are scaled by $\sigma \in [0, 1]$.

**Similarity Metric Contribution Analysis.** We remove a single similarity metric at a time from the full fusion model, which includes all four similarities (WL cosine, TEDS, Const Jaccard, and CM), and compare their retrieval performance (Top-k Recall) on the test set comprising 299 natural number problems. Results show that removing any single similarity metric impacts performance. Notably, removing Const node Jaccard similarity leads to the most significant performance drop, underscoring its critical contribution via semantic overlap measurement, indicating that semantic information remains vital in a structured approach. The main results are summarized in Table 6 .

**Parameter Sensitivity Experiments.** We primarily examine the WL iteration count $h$ and the weights $\alpha, \beta, \gamma, \delta$ used for adaptive similarity fusion (as in Eq. 2). We evaluate the impact of the WL iteration count $h$ on the performance of the preliminary screening stage (Top-k Recall) and the filtering time per query. Experiments on the test set comprising 200 natural number problems indicate that increasing the number of WL iterations improves Recall, but simultaneously significantly increases the filtering time per query. Results reveal a trade-off between accuracy and efficiency, with Recall improvements tending to plateau after a certain number of iterations. Performance and time results for different iteration counts are shown in Table 8. We also examine the impact of the weights $\alpha, \beta, \gamma, \delta$ on the final screening stage performance (Top-k Recall). Analysis on the test set comprising 100 natural number problems indicates that different weight configurations impact retrieval performance. Results show that optimizing weight settings can yield further improvements in Top-k Recall compared to equal weights, confirming the importance of weight tuning for enhancing fusion effectiveness. Experimental results are shown in Table 7.

In addition, we conducted a new experiment specifically analyzing the impact of $\sigma$ on retrieval performance. This parameter controls the scaling of node edit costs in Tree Edit Distance (TED), influencing both precision and efficiency. The results, shown in Table 5, indicate that $\sigma = 0.2$ yields the best performance, with optimal Recall@k. Lower values decrease recall, while higher values significantly reduce it. This highlights the importance of a moderate penalty for simple node edits, enabling TED to better capture structural differences in Lean4 expressions.

Table 5: Impact of $\sigma$

| $\sigma$ Value | Recall@k | |
|---|---|---|
| | @1 | @10 |
| 0.0 | 62 | 67 |
| 0.2 | 67 | 69 |
| 0.4 | 65 | 69 |
| 0.6 | 65 | 69 |
| 0.8 | 61 | 62 |
| 1.0 | 59 | 62 |

Table 6: Accuracy comparison (%)

| Configuration | Recall@k | | |
|---|---|---|---|
| | @1 | @5 | @10 |
| Full Model | 42.8 | 51.1 | 52.5 |
| Without WL | 44.5 | 50.5 | 51.8 |
| Without TEDS | 44.5 | 50.5 | 51.8 |
| Without Jaccard | 37.1 | 42.8 | 44.2 |
| Without CM | 38.2 | 49.3 | 52.4 |

Table 7: Impact of weights on performance (%)

| Weights | | | | Recall@k | | |
|---|---|---|---|---|---|---|
| $\alpha$ | $\beta$ | $\gamma$ | $\delta$ | @1 | @5 | @10 |
| 0.25 | 0.25 | 0.25 | 0.25 | 59 | 63 | 67 |
| 0.50 | 0.20 | 0.20 | 0.10 | 61 | 64 | 68 |
| 0.20 | 0.50 | 0.20 | 0.10 | 59 | 65 | 68 |
| 0.20 | 0.20 | 0.50 | 0.10 | 57 | 63 | 69 |
| 0.10 | 0.20 | 0.20 | 0.50 | 58 | 65 | 68 |
| 0.30 | 0.30 | 0.20 | 0.20 | 59 | 63 | 67 |
| 0.20 | 0.20 | 0.30 | 0.30 | 59 | 64 | 68 |
| 0.15 | 0.40 | 0.30 | 0.15 | 58 | 67 | 70 |

Table 8: WL Iterations vs. Screening accuracy (%,Without TEDS 3.3) and filtering time

| Iterations | Recall@k | | | | | Time (s) |
|---|---|---|---|---|---|---|
| | @100 | @500 | @1000 | @3000 | @5000 | |
| 1 | 54.0 | 66.5 | 71.5 | 77.5 | 81.0 | 20.70 |
| 3 | 55.5 | 67.0 | 73.5 | 80.0 | 81.5 | 27.55 |
| 5 | 57.0 | 68.0 | 73.0 | 79.5 | 82.0 | 32.92 |
| 10 | 55.5 | 65.5 | 70.5 | 78.5 | 83.0 | 47.89 |
| 20 | 55.5 | 66.0 | 70.5 | 78.5 | 82.5 | 55.37 |
| 40 | 56.0 | 65.5 | 70.5 | 78.5 | 83.0 | 54.79 |

## 6 Conclusion

We address the challenge of premise selection in interactive theorem proving systems with large theorem libraries by proposing a novel tree-based method that explicitly leverages the structural information of Lean expressions. Our approach integrates multi-stage filtering, clustering optimization, structural constraints, and adaptive similarity fusion, significantly improving retrieval accuracy. Experiments on the Mathlib4 library show that our method outperforms existing baselines across various Top-k retrieval metrics, demonstrating its superiority for large-scale formal libraries. Furthermore, the framework is designed to be *language-lean*, offering the potential for efficient adaptation to other proof assistants with minimal changes. The inclusion of advanced graph representation techniques and further investigation into Lean expression information are promising directions for future work. Expanding the scale of theorem libraries (e.g., including Optlib), along with tighter integration with automated proof search or LLM proof generation, could provide further advancements in end-to-end proof automation.

## Acknowledgements

This research was supported in part by the National Key Research and Development Program of China (2024YFA1012903), the National Natural Science Foundation of China (12331010 and 12288101), and the Guangdong Provincial Key Laboratory of Mathematical Foundations for Artificial Intelligence (2023B1212010001).

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

# A Data Extraction and Theorem Library Construction

The theorem library used in this study is sourced from Lean4's Mathlib4 version v4.18.0-rc1. An initial extraction yields 339,746 theorems. To construct a high-quality, non-redundant theorem library suitable for the premise selection task, we apply a series of filtering rules to remove theorems that do not meet the requirements, such as internal implementation details, auxiliary lemmas, and theorems from specific namespaces. These filtering rules include excluding theorems from `Lean`, `Std`, `Mathlib.Tactic` namespaces or those whose names contain patterns like `.proof_`, `._auxLemma.`, `.Lemmas.`, or `_private.`. After this filtering process, the final library comprises 217,555 valid mathematical theorems. Each theorem in the library is parsed and converted into an expression tree structure, and stored in a database with key fields including `id` (unique identifier), `name` (theorem name), `statement_str` (formalized statement at Lean4's underlying level), `expr_json` (raw expression tree JSON), `expr_cse_json` (CSE-processed tree JSON), and `node_count` (number of nodes). To assess data quality and understand the characteristics of the theorem library, we conduct statistical analysis on the final set of theorems, focusing on the node count and depth distributions of the theorem expression trees. The node count distribution histogram is shown in Figure 4, and the depth distribution histogram is shown in Figure 5. Statistics on the distribution and structural characteristics of the theorem library across different mathematical domains are detailed in Table 9.

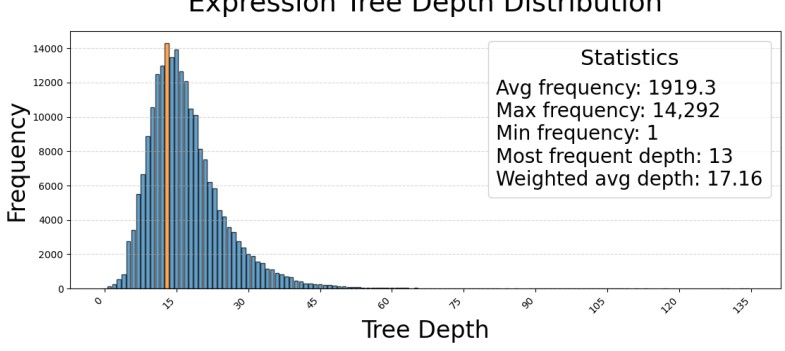

Figure 4: Node Count Distribution Histogram

Figure 5: Depth Distribution Histogram

Table 9: Mathematical Domain Analysis (partial) in `mathlib`

| Domain | Count | Node Count | | | Depth | | | Features |
|---|---|---|---|---|---|---|---|---|
| | | Max | Min | Median | Max | Min | Median | |
| Algebra | 11718 | 1325 | 5 | 95.0 | 130 | 2 | 17.0 | Polynomial/Ring |
| Analysis | 8689 | 1147 | 5 | 125 | 78 | 2 | 20 | Convex/InnerProductSpace |
| Geometry | 1595 | 973 | 11 | 165 | 95 | 4 | 25 | Euclidean/Manifold |
| Topology | 5720 | 819 | 5 | 75.0 | 72 | 2 | 16.0 | Bornology/UniformSpace |
| NumberTheory | 2551 | 986 | 3 | 85 | 59 | 1 | 15 | ClassNumber/Harmonic |
| CategoryTheory | 5570 | 1077 | 5 | 113.0 | 84 | 2 | 20.0 | Adjunction/Functors |
| Logic | 535 | 707 | 3 | 51 | 35 | 1 | 13 | Embedding/Denumerable |

# B   WL Kernel Encoding Process

This section describes the WL kernel method used for encoding expression trees in this paper. The WL kernel is an effective technique for extracting graph structural features, generating compact vector representations that capture the tree's structural properties by iteratively aggregating neighborhood information of nodes. The encoding process can be broken down into the following steps:

1. **Label Initialization (`initialize_labels`):** Initial labels are assigned to each node in the expression tree. The initial label is based on the node's original type; for specific types like `BVar`, `FVar`, `MVar`, `Sort`, `Const`, the label is simplified to the type prefix. Furthermore, each node's label is combined with its depth information in the tree.
2. **Iterative Label Refinement (`wl_iteration`):** $h$ iterations are performed. In each iteration, a node's label is updated based on its label from the previous round and the labels of all its children from the previous round via a hashing function. Specifically, the children's previous labels are collected, sorted lexicographically, concatenated with the current node's previous label to form a string, and finally, the MD5 hash of this string is computed as the node's new label for the current round.
3. **Generate Label Histogram (`get_label_histogram`):** After each iteration, the distribution of all nodes' new labels for the current round is counted to generate a label histogram.
4. **Combine Histograms (`compute_wl_encoding`):** The final tree encoding is generated by combining the label histograms from all $h$ iterations. To distinguish identical labels originating from different iteration rounds, each histogram's labels are prefixed with the corresponding iteration number. The number of iterations $h$ is determined based on the tree's depth and a preset maximum number of iterations, taking the minimum of the two. These combined label counts form the final feature vector used to represent the expression tree.

Through this iterative aggregation process, the WL kernel method is able to capture information about different structural patterns within the expression tree and encode it into a fixed-dimensional feature vector, facilitating subsequent similarity calculations.

# C   Clustering Process

This section describes the process of clustering the WL-encoded theorem vectors. The core objective of clustering is to group theorems with similar structural features to accelerate the preliminary screening speed during subsequent theorem search. We first read the pre-processed theorem trees from the database. These trees are transformed into a more canonical and compact form by applying structural simplification techniques, such as common subexpression elimination (CSE). Subsequently, we apply the Weisfeiler-Lehman (WL) algorithm to encode these simplified trees. These encodings are converted into high-dimensional numerical vectors using a bag-of-features approach, where features correspond to the structural patterns extracted during the WL encoding process. In the feature extraction stage, we select the top 8000 most frequent features as the final feature set by counting feature occurrences across the entire dataset. To measure the distinctiveness of these features and reduce the influence of high-frequency generalized features, we apply an IDF-like weighting scheme to the feature counts in each vector. Considering the high dimensionality of the generated feature vectors, we use Principal Component Analysis (PCA) for dimensionality reduction to mitigate the 'curse of dimensionality' and reduce retrieval computational cost. PCA aims to find

directions of maximum variance in the data, projecting the vectors into a lower-dimensional space comprising 1200 principal components, thereby removing some noise and compressing the data representation. The reduced-dimensional vectors are then clustered using the Mini-Batch K-Means algorithm, an optimized variant of the K-Means algorithm particularly suitable for handling large datasets. Clustering is configured to find 10,000 clusters, utilizing an incremental learning approach where Mini-Batches of 10,000 samples are processed iteratively to update cluster centers. The algorithm converges after a set number of iterations. Upon completion of training, each theorem's vector in the database is assigned the cluster ID corresponding to its nearest learned cluster center, and the results are stored.

The entire clustering pipeline is implemented in Python, primarily utilizing the `PCA` and `MiniBatchKMeans` modules provided by the `scikit-learn` library for core computation, and `psycopg2` for database interaction with the PostgreSQL database. Table 10 presents illustrative examples of clustering results, showing theorems grouped into the same clusters due to structural or semantic similarities, such as the cluster containing `Nat.add_comm` which includes several theorems related to the commutative property of addition. The effectiveness and characteristics of the clustering results are further visualized in Figure 6 and Figure 7. Specifically, Figure 6 presents the sizes of the top-N largest clusters, illustrating the scale of the most populated groups. Complementing this view, Figure 7 shows the histogram of cluster sizes across various predefined bins, providing insight into the overall distribution of cluster memberships, particularly the prevalence of smaller clusters. As is common in clustering diverse datasets, the distribution reveals a substantial number of small clusters, with a rapid decrease in frequency for larger cluster sizes.

Table 10: Display of clustering results for some classical theorems

| Cluster ID 9881
`Nat.add_comm` | Cluster ID 9182
`Nat.add_left_cancel_iff` | Cluster ID 6559
`Nat.eq_zero_of_add_eq_zero_left` |
|---|---|---|
| `Nat.add_comm` | `Nat.add_le_add_iff_left` | `Nat.eq_one_of_mul_eq_one_left` |
| `Nat.add_left_comm` | `Nat.add_left_cancel_iff` | `Nat.eq_zero_of_add_eq_zero_left` |
| `Nat.land_comm` | `Nat.add_lt_add_iff_left` | `Nat.ne_zero_of_mul_ne_zero_right` |
| `Nat.min_comm` | `Nat.add_right_inj` | `AddSubgroup.Normal.conj_mem` |
| `Nat.min_left_comm` | `Nat.xor_right_inj` | `Filter.mem_inf_of_right` |
| `Nat.mul_left_comm` | `BitVec.neg_inj` | `Function.IsPeriodicPt.const_mul` |
| `Bool.or_left_comm` | `BitVec.neg_mul_neg` | `Int.emod_lt_of_pos` |
| `Bool.xor_left_comm` | `BitVec.xor_right_inj` | `List.append_ne_nil_of_ne_nil_right` |
| `Int.max_comm` | `Bool.bne_right_inj` | `List.append_ne_nil_of_right_ne_nil` |
| `Int.min_comm` | `Bool.xor_right_inj` | `List.disjoint_of_disjoint_append_left_right` |

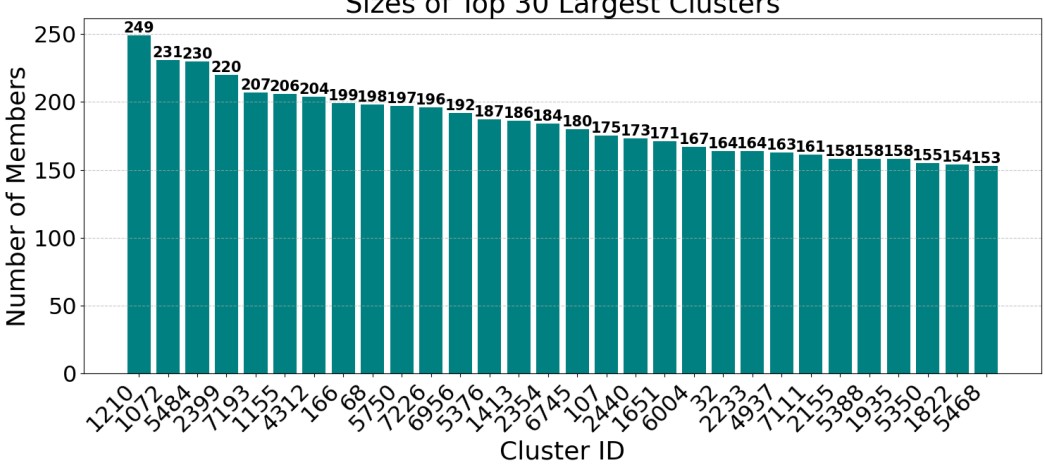

Figure 6: Sizes of the largest clusters (Top N)

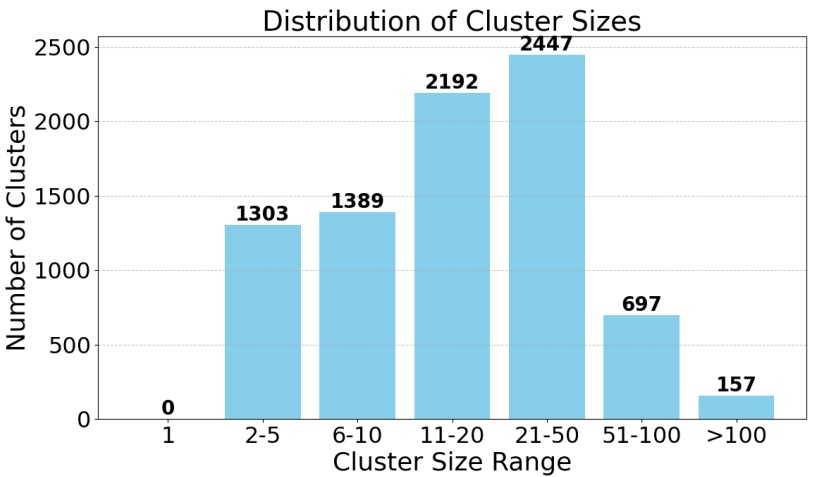

Figure 7: Histogram of cluster size distribution (by bin)

# D    CM Similarity and Jaccard Similarity of `Const` nodes

This section details two additional similarity metrics used in our method: CM similarity and `Const` node Jaccard similarity.

**CM Similarity.** This metric measures how a source expression tree conforms to the structure of a target expression tree based on modified collapse rules. Its core relies on a recursive scoring function: for the current node in the target tree, if it is a leaf node, it contributes 1.0 to the total alignment score regardless of the corresponding node in the source tree; if the current target node is not a leaf, it requires the corresponding node in the source tree to match exactly in label and number of direct children. If they match, the current target node contributes 1.0, and the alignment scores of all corresponding children are recursively computed and summed up; if the label or number of children do not match, this target node and its subtree cannot align, contributing 0. The final CM similarity is the total score obtained by recursively calculating the alignment of the source tree to the target tree structure, normalized by dividing by the total number of nodes in the target expression tree, scaling the similarity value to the $[0, 1]$ range.

**Adjustable Weighting and Application Tendency.** As a crucial metric for evaluating structural conformity, the weight of CM similarity in the final relevance score is adjustable. Users can tune this weight according to their specific proof requirements. For instance, when the objective is to find theorems that can be directly applied (`apply`) to the current proof state, the weight of CM similarity in the overall score can be increased, as this typically signifies a stricter structural match. Conversely, if the goal is to explore broader conceptual associations or find theorems that require multiple transformation steps to be applied, the weight of CM similarity can be reduced, thereby allowing more structurally less-matching but potentially logically relevant theorems to be retrieved. This flexibility enables our premise selection method to adapt to proof tasks of varying complexity and style, offering a more customized search experience.

**Jaccard Similarity of `Const` Node.** This metric aims to capture the semantic overlap between expressions based on the constants used. The calculation process is as follows: Traverse both expression trees and extract the `declName` strings contained within nodes whose labels start with `Const`, collecting them into two sets of unique `declName`s, $S_1$ and $S_2$. Then, compute the Jaccard similarity of these two sets, i.e., $|S_1 \cap S_2|/|S_1 \cup S_2|$. If both sets are empty, the similarity is defined as 1.0. This similarity score reflects the degree to which the two expressions share common constants.

# E  The construction of test sets

## E.1  Test set A

Test set A comprises 100 small-scale problems, designed to evaluate the method's premise selection capability when dealing with theorems exhibiting specific structural variations. These problems are carefully categorized into three types: substitution-type, condition-swapping-type, and mixed-type, respectively assessing the method's ability to recognize structural similarities, adapt to premise reordering, and handle combined transformations. Examples in Lean4 syntax from Test set A are provided in Table 11. Each problem is manually verified to ensure its correctness and alignment with its designated category.

Table 11: Examples from Test Set A: Structured Problem Variants

| Category | Original Theorem | Variant |
|---|---|---|
| Substitution | `theorem Nat.add_comm (n m : Nat) : n + m = m + n` | `example : ∀ (n m : Nat), (n + 1) + m = m + (n + 1)` |
| Condition-swapping | `theorem Nat.nextPowerOfTwo_dec {n power : Nat} (h₁ : power > 0) (h₂ : power < n) : n - power * 2 < n - power` | `example : {n power : Nat} → (h₂ : power < n) → (h₁ : power > 0) → n - power * 2 < n - power` |
| Mixed | `theorem Nat.dvd_gcd {k m n : Nat} : k ∣ m → k ∣ n → k ∣ m.gcd n` | `example : {k m n : Nat} → k.succ ∣ n → k.succ ∣ m → k.succ ∣ m.gcd n` |

## E.2  Test set B

We construct Test set B from `Mathlib4` version 4.18.0 by collecting all `exact` tactic invocations occurring in the proofs of `theorem` declarations. This test set comprises $m = 6{,}119$ (state, theorem) pairs automatically extracted from the Lean proof environment.

For each invocation of the `exact` tactic, we capture the proof state, including local context, current goal, and relevant metadata, immediately before the tactic is applied, and pair it with the theorem supplied to `exact`. We then convert each proof state into a standardized problem statement. We use this dataset to compare our method with existing Lean premise retrieval tools, evaluating their ability to predict the next proof step (the theorem that advances the proof) given a well-defined proof context.

Since the extraction process modifies the Lean compiler, we provide a self-contained Git patch containing the required instrumentation and extraction logic. We have validated this pipeline on multiple Lean4 releases around version 4.18.0, confirming its stability and compatibility across minor revisions.

### E.2.1  Overview of Lean Tactics

In Lean, tactics are high-level commands that guide the proof assistant by transforming the current goal into simpler subgoals or by directly supplying a proof term. For example, the `exact` tactic closes a goal immediately by providing a term whose type matches the goal's statement: `exact h`.

Here, Lean checks that the hypothesis `h` has exactly the required type and uses it to discharge the goal in one step. See Figure 8 for an illustrative flow of how tactics like `exact` operate within a proof.

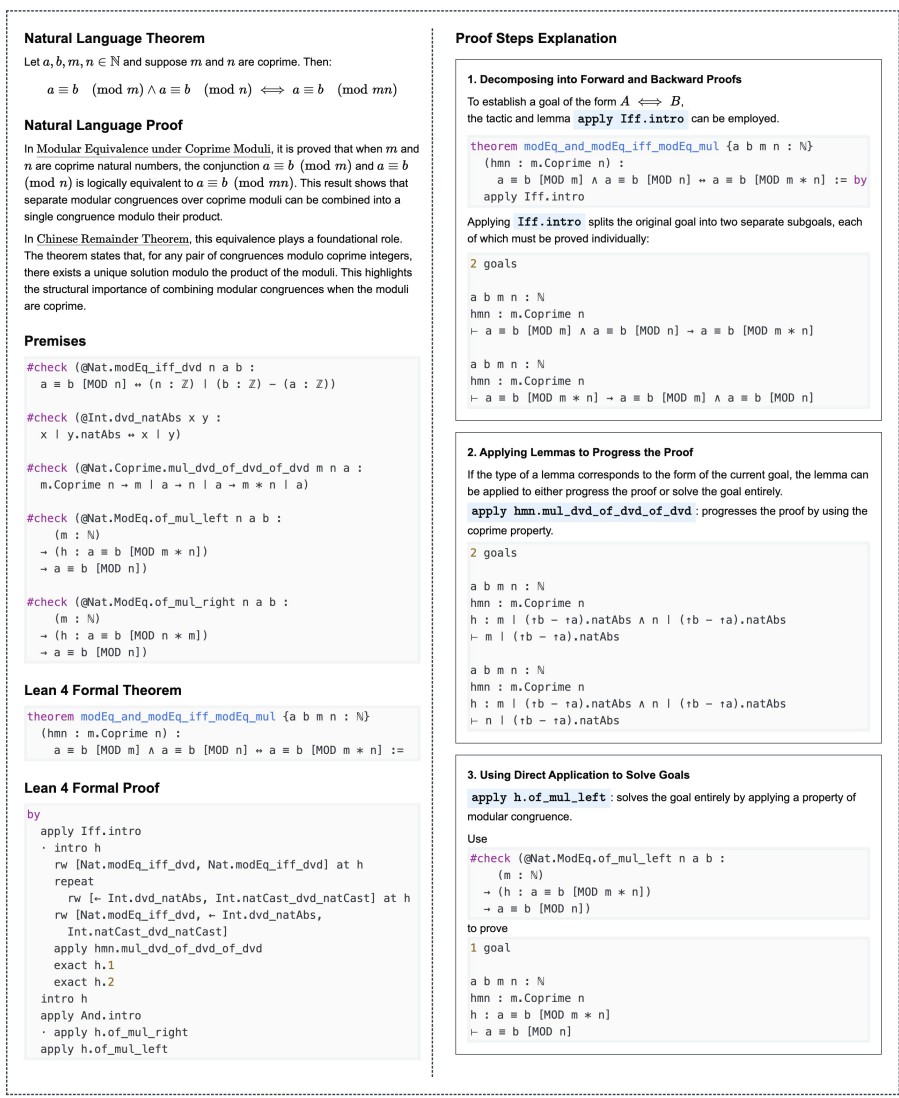

Figure 8: Proof workflow of the `modEq_and_modEq_iff_modEq_mul` lemma in Lean, showing how the two congruence goals are split, rewritten to divisibility, and reassembled under the coprimality assumption.

Furthermore, `exact` is closely tied to premise selection: it requires that the user or an earlier automation has already identified and brought into context a proof object (lemma, hypothesis, or previously constructed term) whose type exactly matches the goal. In practice, this means that successful use of `exact` often follows a search or filtering of available premises to find one that precisely fits, making it the final step in a proof-search or tactic pipeline.

### E.2.2  Extraction and Transformation of `exact` Invocations

In Test set B, we focus on the `exact` tactic: it takes a proof term (typically an existing lemma or theorem) and closes the current goal when the term exactly matches the goal's type. For each invocation of `exact`, the underlying proof step may originate from a combination of higher-order function applications and auxiliary tactics. However, after Lean's elaboration phase, every such invocation is represented as an `Expr` of the form $(f\,x\,y\,z\,\ldots)$ where $f$ is the theorem being applied and $x, y, z, \ldots$ are its explicit arguments. In our Test set B generation pipeline, we collect the inferred types of all these explicit arguments and append them to the original goal, thereby yielding a fully instantiated goal proposition with a complete type signature that reflects the concrete instantiation scenarios encountered in premise selection. The transformation illustrated in Figure 9 exemplifies

how our extraction pipeline turns a raw `exact` invocation into a fully instantiated search query, by appending every explicit argument type to the original goal.

```
exact Nat.pow_le_pow_left
    (add_le_add_right (Nat.mul_div_le i j) _) -- explicit argument 1
    (_) -- explicit argument 2:
        -- underscores '_' are synthesized by Lean's elaborator via type inference.
```

(a) Paired theorem invocation

```
▼Tactic state
1 goal
i j : ℕ
hj : 0 < j
this : j * (j − 1) < j ^ 2
⊢ (j * (i / j).succ) ^ 2 ≤ (i + j) ^ 2
```

```
▼Tactic state
1 goal
i j : ℕ
hj : 0 < j
this : j * (j − 1) < j ^ 2
⊢ j * (i / j) + j ≤ i + j → ∀ (x1 : ℕ),
  (j * (i / j).succ) ^ 2 ≤ (i + j) ^ 2
```

(b) Captured proof state

(c) Transformed proof state

Figure 9: Example of transformation of a proof state under an `exact` invocation in Test set B. (a) displays the corresponding `exact` tactic invocation using a composed proof term. (b) shows the proof state immediately before the tactic is applied. (c) shows the reconstructed, logically equivalent but more structured goal that exposes the necessary conditions for the theorem application.

In our pipeline, each transformation as illustrated in Figure 9 is recorded and serialized as JSON. This machine-readable format makes it easy to generate valid search queries for different premise retrieval tools, allowing systematic testing of our method and other approaches.

This transformation is crucial for evaluating premise-retrieval tools: although the constant $f$ often has a very general polymorphic type, in real proof developments the goals resolved by `exact` are highly instantiated and structurally complex. By exposing the concrete argument-type information, our dataset rigorously tests a tool's ability to select the correct theorem under realistic, nontrivial instantiations.

### E.2.3 Implementation Details of the Data Transformation and Extraction Pipeline

To support this extraction, Lean tactics operate within the `TacticM` monad, which provides access to:

- **Local context**: the list of hypotheses and their types;
- **Goal**: the current proposition to be proved;
- **Metadata**: such as proof position, the name of the invoking lemma or theorem, and environment declarations;
- **Elaboration**: converting `Syntax` objects into fully elaborated `Expr` terms;
- **Meta-variable resolution**: instantiating and solving meta-variables generated during proof construction;
- **Type inference**: inferring the type of arbitrary `Expr` terms within the current environment.

To construct the test set, we instrument the built-in `exact` tactic to capture structured information at each invocation. The extraction process operates as follows.

1. We extend the `TacticM` environment and context to record additional state required for the transformation with additional internal state, including elaboration results, meta-variable assignments, and relevant `Syntax` information required during reconstruction.

2. When `exact e` is executed, we first elaborate the term $e$ to an expression whose type must match the current goal. We then attempt to extract the head constant name (i.e., the applied theorem) from the syntax and expression of $e$, using a combination of syntactic resolution and application spine inspection.

3. Once the theorem is identified, we retrieve its type, instantiate any meta-variables in the argument terms, and collect the types of all explicitly passed arguments. We then append these instantiated argument types to the current goal, producing a normalized target type that captures both the proof context and the specific theorem application, while preserving the original proof state unmodified.

4. We compute a simplified version of the goal by recursively reducing all subexpressions. This normalization step helps eliminate intermediate constructs introduced by instance resolution and other elaboration mechanisms, making the goal type easier to analyze and compare.

5. In addition to the core type-level information, we also collect auxiliary metadata such as the source file URI, position range, command kind, and a reprint of the original tactic syntax. All extracted information, including the normalized goal type and associated context, is serialized into a JSON object, which is then uploaded to a PostgreSQL database for downstream consumption.

# F   Detailed Visualization of Search Results

This section provides detailed visualizations of search outcomes to further elucidate the efficacy and operational mechanism of the proposed premise selection methodology. Examples of query expressions and their corresponding matched theorems retrieved by the system are visually depicted as tree structures in Table 13. These examples, derived from variants of theorems like `Nat.add_comm`, allow for a qualitative inspection of the structural similarities leveraged by the method for determining relevance. Through these successful retrieval cases, the immense power of structural information in identifying pertinent theorems is clearly demonstrated. Further concrete examples of premise retrieval are detailed in Table 12, listing target expressions in Lean format alongside the names of the top-5 theorems retrieved by the system. Collectively, these figures and tables offer readers detailed qualitative and concrete examples for a deeper understanding of the search process's operational characteristics and the underlying principles of our method.

Table 12: Illustrative examples of retrieved theorems for selected target expressions.

**Target Expression in Lean:**
(n m : Nat) → (n + 1) + m = m + (n + 1)

**Top-5 Retrieved Theorem(s) (Name):**
Nat.max_comm; Nat.add_comm; Nat.min_comm; Nat.xor_comm; Nat.or_comm;

**Target Expression in Lean:**
{a b c : Nat} → (h : a + c <b) → ¬ b <a + c

**Top-5 Retrieved Theorem(s) (Name):**
Nat.ModEq.symm; Batteries.UnionFind.Equiv.symm; Nat.lt_asymm; Nat.ModEq.comm;
Nat.le_total;

**Target Expression in Lean:**
{b a : Nat} → ($h_1$ : 0 <b) → ($h_2$ : b ≤ a.succ) → a.succ / b = (a.succ - b) / b + 1

**Top-5 Retrieved Theorem(s) (Name):**
Nat.div_eq_sub_div; Nat.choose_le_succ_of_lt_half_left;
Nat.le_two_mul_of_factorization_centralBinom_pos; Nat.log_mul_base;

**Target Expression in Lean:**
(m n k : Nat) → (m.succ * n).gcd (m.succ * k) = m.succ * n.gcd k

**Top-5 Retrieved Theorem(s) (Name):**
Nat.gcd_mul_left; Int.gcd_natCast_natCast; Nat.gcd_assoc; Nat.lcm_mul_left; Nat.dvd_gcd;

**Target Expression in Lean:**
{n : Nat} → {a b : Fin n.succ} → (h : a <b) → ↑a + 1 ≤ ↑b

**Top-5 Retrieved Theorem(s) (Name):**
Fin.add_one_le_of_lt; Fin.cycleRange_of_gt; Fin.add_one_lt_iff; Fin.add_one_le_iff;
lt_finRotate_iff_ne_last;

Table 13: Visual Trees of Proposition Search Results

| Query Proposition Tree | Matched Theorem Trees |
|---|---|

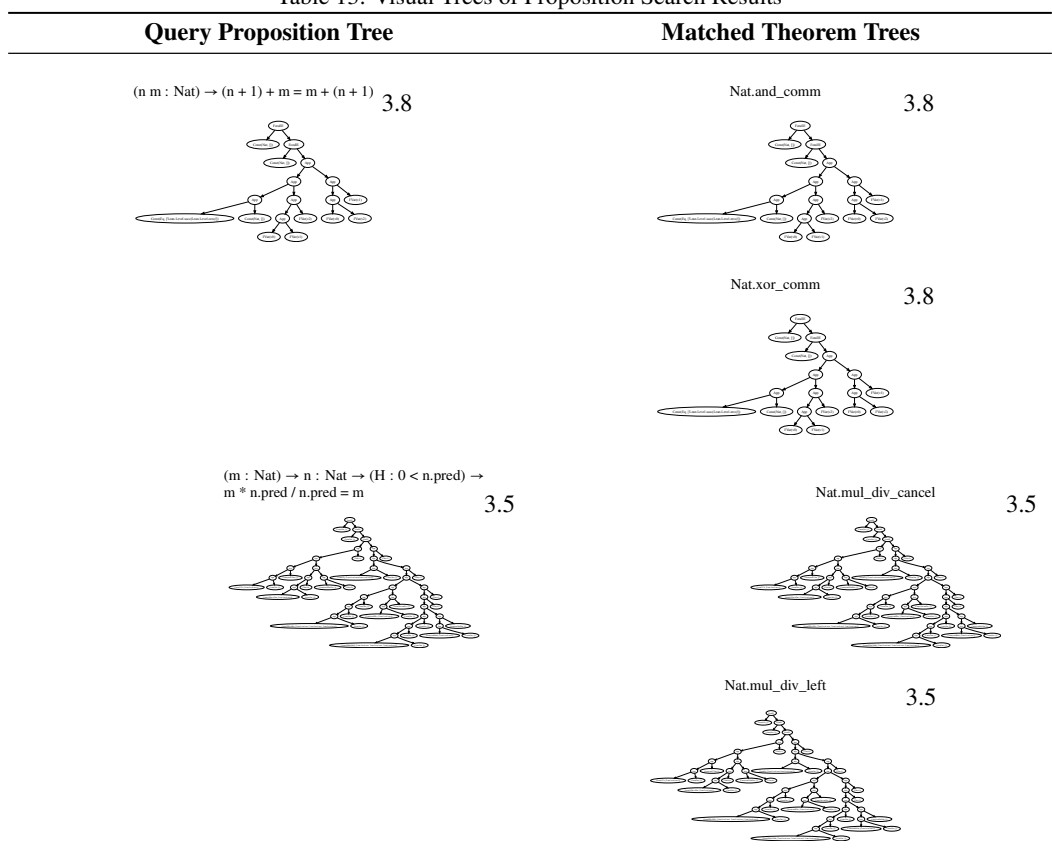

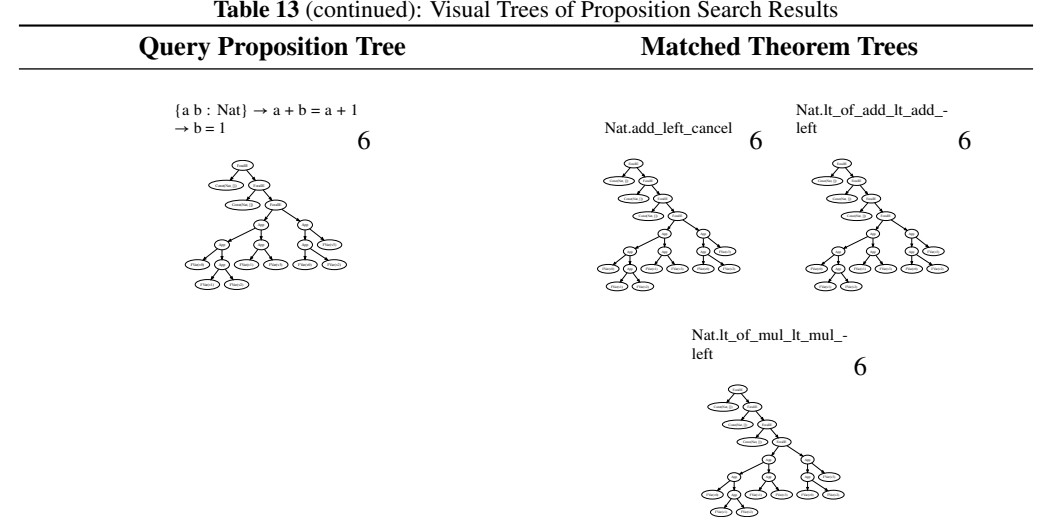

## G Limitations

Despite the significant effectiveness achieved by the proposed method in premise selection, certain limitations remain that can be addressed in future work. First, complex definition unfolding and implicit instance conversions in Lean can lead to expressions with similar mathematical meaning exhibiting considerable structural differences at the tree level. This structural divergence increases the difficulty of precise matching, necessitating further research into unifying or more robustly handling these structural variations. Second, there is still room for speed optimization. Current experiments are primarily conducted on local computing resources; on more powerful machines, retrieval efficiency can be further improved through enhanced parallelization and database-level optimizations (e.g., placing hot data in an in-memory database).

A core challenge stems from the inherent complexity of Lean 4 expression trees and the dynamic nature of its proof environment. An illustrative example involves the query expression `padicValNat p n < padicValNat p n + 1` (depicted in Figure 12) and its relevant theorems such as `Nat.lt_-succ_self: n < n.succ` (shown in Figure 10) and `Nat.lt_add_one: n < n + 1` (illustrated in Figure 11). While `n.succ` and `n + 1` are mathematically equivalent, Lean's automatic implicit instance conversion for `+ 1` results in structurally different expression trees. Lean's `exact` tactic can automatically unfold definitions to handle such equivalences during theorem application, our method, by strictly adhering to the expression tree structure, perceives `n.succ` (representing the definition of successor) and `n + 1` as distinct sub-tree structures. This structural distinction, as evident when comparing Figure 10 and Figure 11, means that even theorems that are semantically identical to the query can exhibit significant structural differences in their corresponding expression trees.

Furthermore, Lean's automatic implicit instance conversion mechanism poses another significant challenge for tree-based structural matching. For instance, what might appear as a simple `Const(instLTNat, [])` node (representing a type class instance) in a theorem's simplified representation could expand into a large, complex instance conversion sub-tree within the target expression due to type inference and instance lookup. This structural inflation is particularly noticeable when observing the second node (or its corresponding sub-tree) from the left at the fourth level of the expression trees, as depicted in Figure 12 and Figure 11. These "large and complex instance conversion sub-trees" automatically inserted by the compiler and not directly related to the core mathematical semantics, drastically increase the structural discrepancy between query and target expression trees. Both the definition unfolding (e.g., `n.succ` vs. `n + 1`) and the implicit instance conversions contribute to these structural divergences, consequently impacting the retrieval accuracy of tree similarity algorithms. While our CM similarity metric was designed to partially mitigate this by introducing a match-degree based score rather than strict structural equality to enhance robustness, the overall structural inflation remains a critical area for further research.

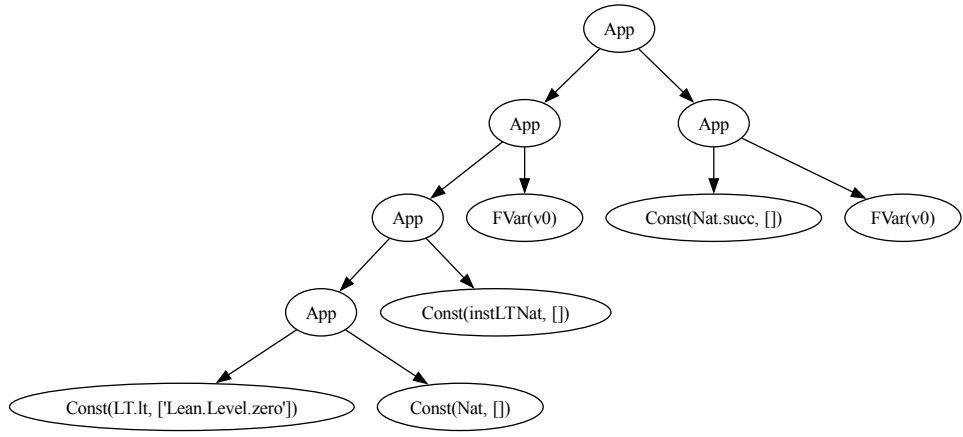

Figure 10: Simplified expression tree of theorem `Nat.lt_succ_self`

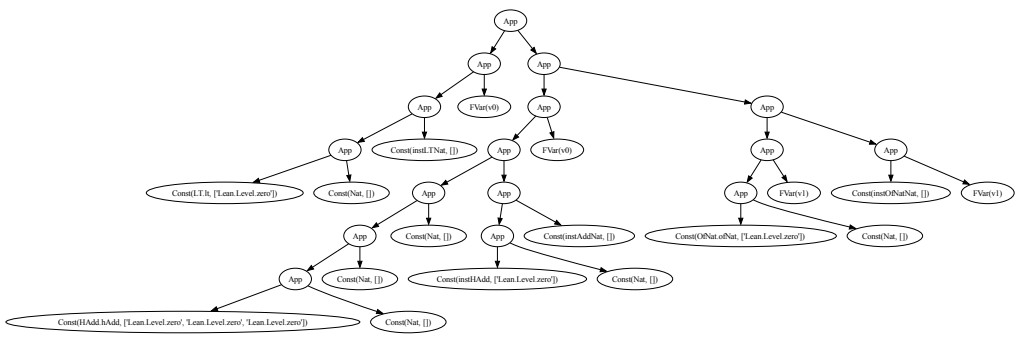

Figure 11: Simplified expression tree of theorem `Nat.lt_add_one`

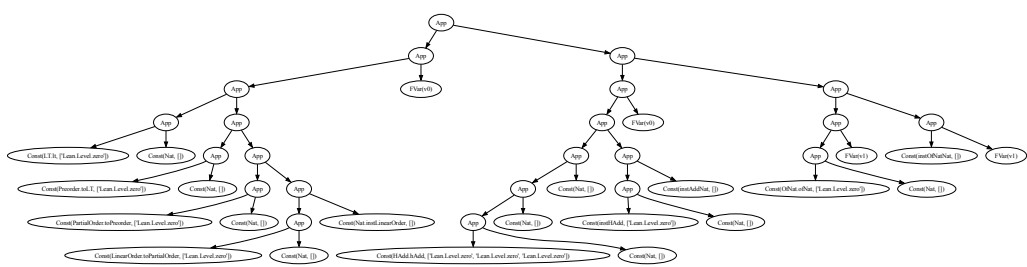

Figure 12: Simplified expression tree of query `padicValNat p n < padicValNat p n + 1`

Future work will focus on exploring more intelligent expression tree normalization techniques, such as deeper $\alpha$-equivalence or $\beta$-reduction prior to WL encoding, or developing tree similarity algorithms capable of recognizing and ignoring compiler-inserted structures (like implicit instance conversions) that do not affect the core mathematical semantics.

# H    Broader Impacts

The premise selection method proposed in this paper has significant broader impacts on the fields of interactive theorem proving and automated theorem proving. By increasing the efficiency and accuracy of retrieving relevant premises from large formal theorem libraries, this work directly addresses a critical bottleneck in automating proofs and formalizing human proofs. This can help accelerate the automation of complex mathematical proofs, facilitate the application of formal verification in software, hardware, and safety-critical systems, thereby enhancing their reliability and security. Furthermore, improved premise selection tools can assist mathematicians in more conveniently formalizing their proofs, exploring mathematical structures, and potentially fostering new mathematical research.

However, this work also entails potential ethical considerations and impacts. For instance, over-reliance on such automated tools might diminish human provers' ability to independently discover and understand proof steps. Additionally, the outcomes of premise selection may be influenced by the existing formalization styles and biases present in the theorem libraries and training data used, potentially unintentionally favoring certain proof approaches or forms of expression. Future research and tool development should focus on ensuring the transparency and interpretability of these tools, encouraging users to maintain critical thinking, and actively addressing data bias issues to ensure technological advancements serve broader scientific and educational goals.

# I Extended Performance Evaluation

## I.1 Analysis of Efficiency and Performance Trade-off

This section focuses on analyzing the trade-off between efficiency and performance of the proposed clustering-based premise selection method. To quantify the impact of the clustering mechanism, we compare the full method ("With Clustering") against a baseline of full-library search without preliminary clustering ("Without Clustering"). All experiments were conducted on test set A, recording the query time distribution (Min, Average, Max), total time, speedup, and Recall@k at various k values for each method.

Table 14 presents a detailed comparison of the results. Regarding query time, the "With Clustering" method demonstrates a significant advantage in average query time. Its average query time of 14.574 seconds is substantially lower than the 30.973 seconds of the "Without Clustering" method. This translates to a remarkable speedup of **2.13x**, indicating that the clustering preliminary screening mechanism successfully narrows down the search space, thereby significantly boosting retrieval efficiency. Even in the worst-case scenario (Max Query Time), the "With Clustering" method exhibits superior performance (99.011 seconds vs. 110.329 seconds). However, this gain in efficiency comes with a certain degree of accuracy trade-off. From the Recall@k metrics, the "With Clustering" method shows a slight decrease in Recall@1, Recall@5, and Recall@10 compared to the "Without Clustering" baseline. For instance, Recall@10 drops from 95.0% to 86.0%. This suggests that clustering, as an approximate filtering method, might filter out a small number of truly relevant theorems that were not assigned to the target clusters, in exchange for accelerated retrieval. Despite the marginal decrease in recall, we argue that this efficiency-performance trade-off is highly practical and acceptable for interactive theorem proving environments. In the context of real-world theorem proving, users typically prioritize quickly obtaining a high-quality and explorable set of preliminary results to promptly proceed with their proof attempts or strategy adjustments, rather than spending extensive time waiting for a precise full-library ranking. The 2.13x speedup can significantly enhance user experience and improve the iterative efficiency of the entire proving process. This study successfully demonstrates that by incorporating structured representations and efficient clustering for preliminary screening, a qualitative leap in theorem retrieval efficiency can be achieved while maintaining sufficient accuracy.

Table 14: Significant Efficiency Gains of Clustering-based Premise Selection

| Method | Query Time (s) | | | Total Time (s) | Speedup | Recall@k (%) | | |
|---|---|---|---|---|---|---|---|---|
| | Min | Average | Max | | | @1 | @5 | @10 |
| Without Clustering | 2.295 | 30.973 | 110.329 | 3097.3 | 1.00x | 90.0 | 91.0 | 95.0 |
| With Clustering | 1.760 | 14.574 | 99.011 | 1457.4 | **2.13x** | 82.0 | 84.0 | 86.0 |

## I.2 Domain-Specific Performance Comparison

To comprehensively evaluate our proposed "Tree-Based Search" method, a comparison was conducted against several existing theorem retrieval approaches (including Lean Search, Moogle, Lean Search Agu, Lean Explore, and Lean State Search) within a specific mathematical domain. This experiment focused on the Natural Number domain, with theorems further categorized by complexity into "Simple," "Medium," and "Complex" levels. The evaluation metric remained Recall@k (k=1, 5, 10), aiming to measure the system's ability to recall relevant theorems at various retrieval depths.

The retrieval performance across different complexity levels within the Natural Number domain is summarized in Table 15. From the results, it is evident that our "Tree-Based Search" method consistently and significantly outperforms all compared methods across all complexity levels. Notably, for both "Simple" and "Medium" complexity Natural Number theorems, the highest Recall@1, Recall@5, and Recall@10 were achieved by our method, with Recall@10 reaching 82.7% in the "Simple" category, far exceeding the next best approach.

As theorem complexity increases, a general decrease in retrieval performance is observed across all methods. However, even in the most challenging "Complex" category, a relatively significant advantage is maintained by "Tree-Based Search" (Recall@10 reaching 38.1%), whereas other methods

often show recall rates near or at 0%. This strongly demonstrates the robustness and effectiveness of our tree-based structural representation and similarity matching approach in handling mathematical expressions of varying complexity, especially in more complex theorems where its ability to capture deeper structural information becomes crucial.

Table 15: Retrieval performance (%) of different methods on natural number domain categorized by complexity

| Domain | Complexity | Count | Method | Recall@k | | |
|---|---|---|---|---|---|---|
| | | | | @1 | @5 | @10 |
| Natural Number | Simple | 76 | Lean Search | 17.1 | 26.3 | 27.6 |
| | | | Moogle | 7.9 | 14.5 | 22.4 |
| | | | Lean Search Agu | 16.0 | 20.0 | 20.0 |
| | | | Lean Explore | 2.7 | 20.0 | 38.7 |
| | | | Lean State Search | 30.3 | 50.0 | 63.2 |
| | | | **Tree-Based Search** | **68.0** | **81.3** | **82.7** |
| | Medium | 126 | Lean Search | 6.3 | 15.9 | 21.4 |
| | | | Moogle | 0.0 | 4.8 | 7.9 |
| | | | Lean Search Agu | 7.2 | 15.2 | 20.0 |
| | | | Lean Explore | 0.0 | 2.4 | 9.5 |
| | | | Lean State Search | 10.3 | 23.8 | 25.4 |
| | | | **Tree-Based Search** | **38.9** | **42.1** | **44.4** |
| | Complex | 21 | Lean Search | 0.0 | 0.0 | 0.0 |
| | | | Moogle | 0.0 | 4.8 | 4.8 |
| | | | Lean Search Agu | 0.0 | 0.0 | 9.5 |
| | | | Lean Explore | 0.0 | 0.0 | 0.0 |
| | | | Lean State Search | 4.8 | 14.3 | 14.3 |
| | | | **Tree-Based Search** | **23.8** | **38.1** | **38.1** |

This detailed performance is further illuminated through various visualizations. A comprehensive 4x4 comparison of our method against all other models across 'Recall@1', 'Recall@5', 'Recall@10', and 'MRR' metrics within the Natural Number domain is provided in Figure 14, clearly demonstrating our consistent lead. The distribution of search queries by difficulty level in this domain is presented in Figure 13, offering context on the dataset composition. Furthermore, the trends of key metrics (MRR, Recall@5, nDCG@10) across different difficulty levels are visualized in Figure 15, highlighting our method's robust performance even as complexity increases. A focused view on the '@1' metrics (Recall@1, Precision@1, F1@1, and nDCG@1) is provided in Figure 16, offering granular insights into the initial retrieval effectiveness. Collectively, these visualizations underscore the superior effectiveness of our Tree-Based Search method across different evaluation criteria and complexity levels.

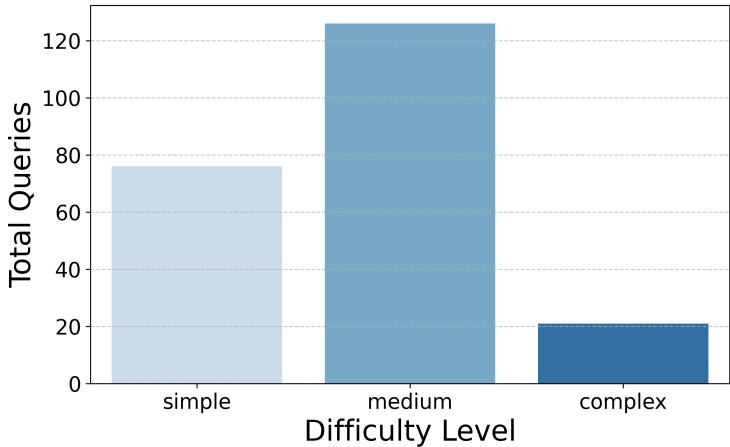

Figure 13: Total Queries by Difficulty Level in Natural Number Category.

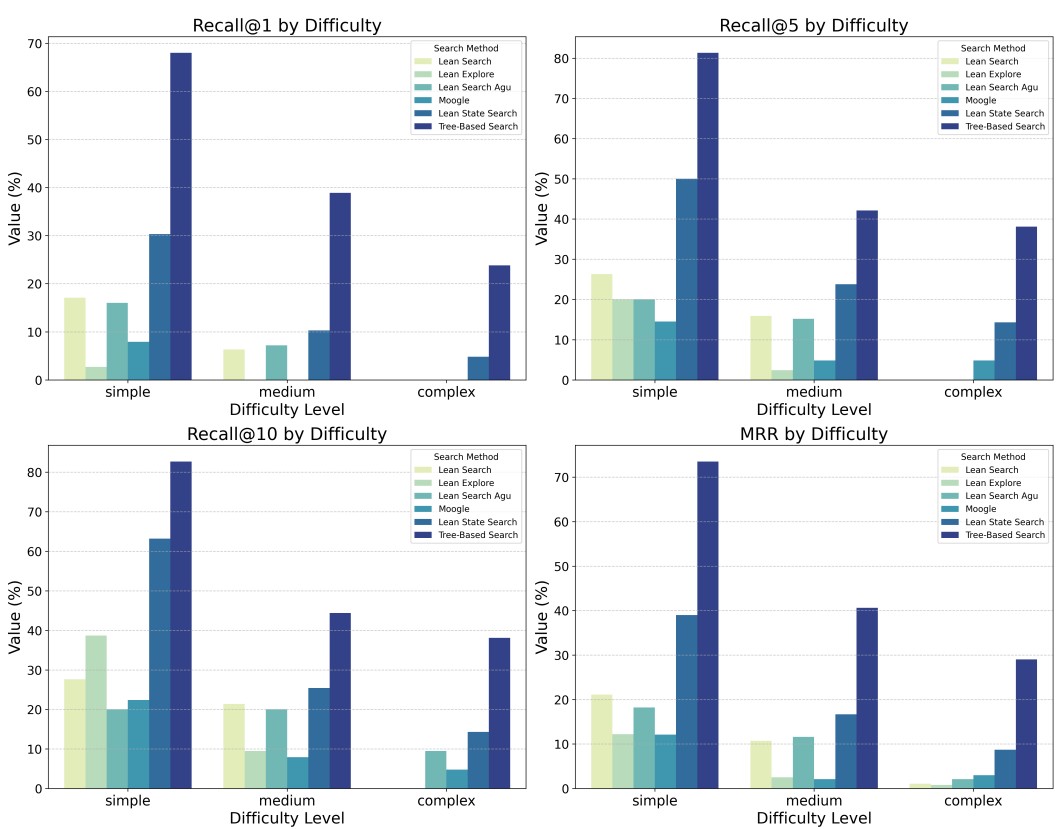

Figure 14: Performance comparison of search methods across Recall@k and MRR by difficulty level in Natural Number domain.

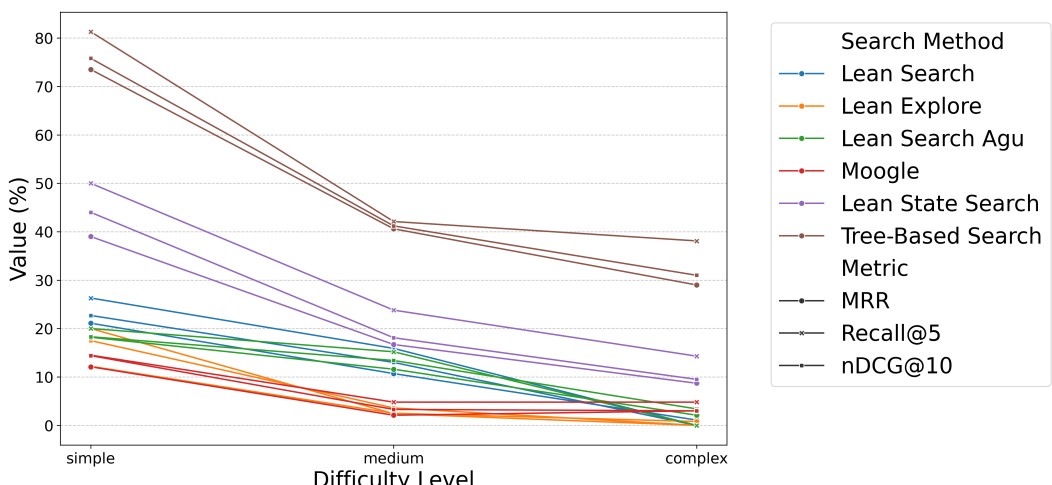

Figure 15: Key Metric Trends (MRR, Recall@5, nDCG@10) by Difficulty Level in Natural Number Category.

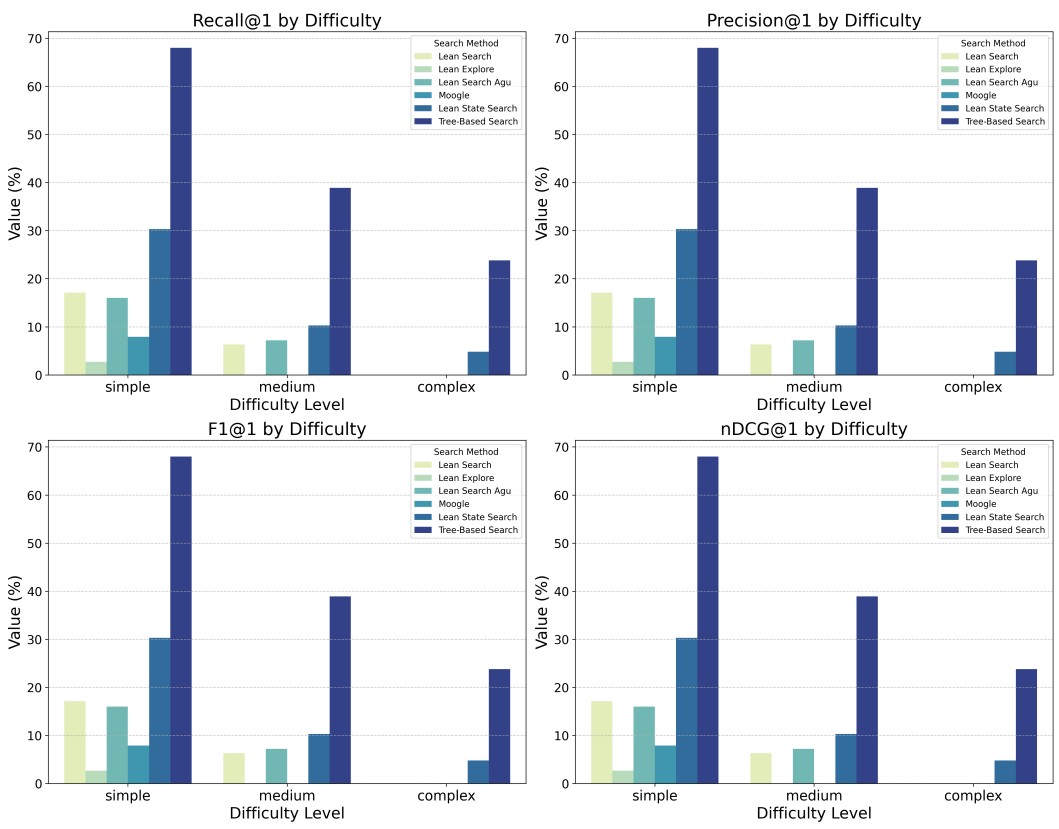

Figure 16: Grouped Bar Charts for @1 Metrics (Recall@1, Precision@1, F1@1, nDCG@1) by Difficulty Level.

