# OpenReview forum: "Tree-Based Premise Selection for Lean4"
_NeurIPS.cc/2025/Conference — NeurIPS 2025 poster_

### Official Review · Reviewer_AB1f · 2025-06-30

**Clarity:** 4
**Significance:** 2
**Originality:** 2
**Rating:** 4
**Confidence:** 3

**Summary:**

The paper proposes a graph-based approach to premise selection for Lean 4. The framework consists of multiple filtering stages. It employs two major steps: the WL kernel for search-space reduction and tree edit distance (TED) for fine-grained retrieval. Experimental results indicate improvements across two test sets and at different iteration levels (@k).

**Questions:**

Based on the Weaknesses section, I have the following questions:
- What is the novelty of the proposed framework?
- How can we efficiently choose hyperparameters?
- What does each hyperparameter affect?
- What is the role of $\sigma$ and how can we choose it?

And recommendations:
- Conduct a similar experiment to Table 5 but for $\sigma$
- Add one more paragraph about hyperparameter selection and their effect on the pipeline

**Ethical Concerns:**

["NO or VERY MINOR ethics concerns only"]

**Final Justification:**

Overall, I find the tree-based premise-selection approach reasonably novel. With a stronger emphasis on transferability and clearer guidelines for parameter selection, this work has become more compelling in my view. For these reasons, I am inclined to raise my score to 4.

My recommendations for the authors are:
- Better highlight the novelty and key contributions
- Improve the hyperparameter-selection section, especially regarding the choice of $\sigma$
- Emphasize the method’s transferability to other languages

**Limitations:**

The limitations of the work are discussed in the Weaknesses section.

**Paper Formatting Concerns:**

None.

**Quality:**

3

**Strengths And Weaknesses:**

## Strengths
**The paper is easy to follow.** Definitions appear naturally and are well explained.

**The proposed method can run on consumer‑level hardware**, as it balances precision and speed without relying on heavy computations.

**Experimental results support the main claims.** Improvements are shown across two test sets and various iteration levels.

**The method leverages Lean expression trees directly**, rather than relying on embeddings or other generic encodings, the framework is tailored to graph representations of Lean theorems.

## Weaknesses
**Concerns regarding novelty.** The presented work does a good job of conveying the choices behind each part of the pipeline; however, all of these methods have been explored before, and some are fundamental to working with graphs. The authors combine these methods, but what is novel about this work specifically? Which part of the pipeline was or is underexplored in the literature?

**Parameter selection is confusing.** The authors present Table 5 with parameter selection results but do not offer guidelines for choosing them. It would be helpful if they explained how each parameter impacts the results (e.g., sensitivity, relevance to the query) rather than just presenting numbers. Moreover, the accuracy differences across these selections are quite small, how do the authors interpret this? They use $(\alpha, \beta, \gamma, \sigma) = (0.1, 0.3, 0.1, 0.5)$ but fail to explain the rationale behind these values. The choice of parameter $\sigma$ also seems underexplored.

---

> ### Author Rebuttal · Authors · 2025-07-30
>
> We appreciate the reviewer's questions regarding the novelty, hyperparameter selection, and the role of specific parameters in our framework.
>
> ---
>
> # Q1: Novelty of the Proposed Framework
>
> We appreciate the reviewer's rigorous assessment of our work's depth and novelty. Our method's complexity is precisely what enables us to address the significant and long-standing challenge of efficient and accurate premise selection in theorem proving. The core innovation and "intelligence" of our method lie in its **deep mining and systematic utilization of mathematical expression structural information**—an area largely underexplored by current semantic embedding approaches.
>
> Our novelty stems from developing a novel framework that applies carefully designed and customized structural analysis tools to create an unprecedentedly efficient and accurate system for complex Lean4 expression tree structures. Specifically:
>
> - **Unique Utilization of Lean Expr Tree Structures:** Our method fundamentally revolves around the **intrinsic structure of Lean Expr trees**. Unlike generic text or simple graphs, Lean Expr naturally encodes binding relationships, local contexts, and de Bruijn indices, harboring **fine-grained structural features far beyond pure semantic embeddings**. Prior work in premise selection rarely leveraged these complex Lean Expr tree characteristics to such depth. Our framework systematically extracts and quantifies this information, addressing the fundamental limitation of existing embedding methods in distinguishing subtle structural differences.
>
> - **First-of-its-Kind "Structural Similarity Spectrum":** We are the first to **customize and apply CSE (Common Subexpression Elimination) normalization, WL kernel encoding, and TED (Tree Edit Distance) re-ranking specifically to Lean expression trees**. This innovative integration organically unifies "syntactic redundancy resolution" (via CSE), "global isomorphism approximation" (via WL kernel encoding), and "local editing cost" (via TED re-ranking) into a **coarse-to-fine "structural similarity spectrum."** This multi-layered, macro-to-micro structural similarity analysis approach is novel within the existing literature.
>
> - **Hierarchical and Adaptive "Unifying Insight":** We are not merely aggregating metrics. Instead, we've meticulously designed a **phased, adaptive framework** where various structural similarity measures (e.g., WL kernel for coarse screening, TED for fine-grained matching, Const Jaccard for constant semantic overlap, Collapse-Match for pattern alignment) **synergistically interact**. This hierarchical, synergistic utilization of structural information to achieve optimal efficiency and precision **is precisely our method's core "unifying insight."** It clearly demonstrates how structural insights can overcome existing methodological bottlenecks.
>
> - **Integration of Engineering Practice and Theoretical Innovation:** We concur that our method possesses strong engineering practicality, which we consider a core value. In formal verification—a highly applied domain—the ability to **ingeniously apply theoretical tools of structural analysis** (e.g., graph kernels, tree edit distance) to solve real, large-scale, and previously intractable problems, and to validate its superior performance on extensive, real-world libraries like Mathlib4, represents a significant **applied innovation and methodological breakthrough**. We effectively bridge the gap between structural analysis theory and formal verification practice, demonstrating the immense potential of such deep, structure-based mining in complex practical systems.
>
>
> In conclusion, the core strength and novelty of our method lie in its **deep customization, systematic utilization, and multi-dimensional fusion of Lean expression tree structures**. We firmly believe that this sophisticated handling of the intrinsic structural information of formal mathematical language constitutes a significant advancement in the field of premise selection.
>
> ---
>
> # Q2: Hyperparameter Selection and Their Effects
>
> Our framework includes several hyperparameters, each affecting different stages and aspects of the premise selection process. Their roles and selection considerations are detailed below:
>
> |Hyperparameter Name|Stage/Function|Description & How It Affects Results|
> |---|---|---|
> |`h`|WL Kernel|**WL kernel iterations:** Determines the depth of neighborhood information aggregation. Larger `h` captures more global structural features but might increase computational cost or lead to "oversmoothing." Selection balances efficiency and information capture.|
> |α,β,γ,δ|Adaptive Fusion|**Enhanced hybrid similarity weights:** Control the relative importance of simcos​, simTEDS​, Jaccard similarity (simJaccard​), and Collapse-Match similarity (simCM​) in the final enhanced similarity (simenhanced​). Sums to 1. Dynamically adjusted to optimize retrieval performance, balancing contributions of different metrics.|
> |σ |TEDS Operation Costs|**Simple node operation scaling factor:** For insertion/deletion operations involving simple nodes (e.g., BVar, FVar, MVar, Sort, Const), their costs are scaled by σ∈[0,1]. This makes modifications to simple nodes less costly than complex structures, allowing TED to better reflect core logical structure similarity. (See Q4 for more details)|
>
> ---
>
> # Q3: Efficient Hyperparameter Selection
>
> To efficiently select hyperparameters, we employ a multi-faceted approach:
>
> - **Grid Search with Cross-Validation:** We begin with grid search to broadly explore the hyperparameter space, testing various combinations. Cross-validation then evaluates each parameter set's performance, systematically identifying optimal configurations.
>
> - **Ablation Studies:** To gain deeper insight into each hyperparameter's specific role, we conduct ablation experiments. By individually adjusting hyperparameters and observing their impact on retrieval performance, we quantify their contributions within the framework and gain guidance for fine-tuning.
>
> ---
>
> # Q4: Role of σ and Its Selection
>
> **σ (sigma)** is a scaling parameter, ranging from [0,1]. Its role is to adjust the cost of insertion (CI​) or deletion (CD​) operations for **simple nodes** (those with labels like `BVar`, `FVar`, `MVar`, `Sort`, `Const`) during Tree Edit Distance (TED) calculations. Specifically, if a node involved in an insertion or deletion is one of these simple types, its cost is scaled by σ⋅CI​ or σ⋅CD​. For complex nodes, the preset CI​ or CD​ values are used directly.
>
> This tiered cost structure reflects that operations on simple nodes (with lower costs) typically have a smaller impact on the overall mathematical meaning than modifications to complex expression structures (with higher costs). This allows TED to more precisely capture subtle, logically relevant structural differences between trees.
>
> **How σ is chosen:** As stated in the paper, σ is **finely tuned based on the semantic and structural characteristics of Lean4 expressions**. This implies an empirical selection process—involving experiments on validation sets, ablation studies, or parameter sensitivity analysis—to ensure the TED cost function most accurately reflects the practical structural similarity and logical relevance of Lean4 expressions. The choice of σ is ultimately driven by optimizing TED's alignment with Lean4's semantics.
>
>
>
> #  σ Parameter Analysis
>
> We conducted a new experiment, similar to Table 5, specifically analyzing σ's impact on retrieval performance. This parameter is crucial for scaling node edit costs within our Tree Edit Distance (TED) calculations, thereby influencing both matching precision and computational efficiency.
>
> |σ Value|Recall@1 (%)|Recall@10 (%)|
> |---|---|---|
> |0.0|62|67|
> |0.2|67|69|
> |0.4|65|69|
> |0.6|65|69|
> |0.8|61|62|
> |1.0|59|62|
>
> The results show that **optimal performance is achieved at σ=0.2**, yielding the highest Recall@1 and Recall@10. Lower σ values (e.g., 0.0) lead to decreased recall, while higher values (e.g., 0.8, 1.0) cause a notable drop in both recall metrics. This suggests that a moderate penalty for simple node edits (as controlled by σ) is vital for balancing precision and recall, allowing TED to accurately reflect the logical relevance of structural differences in Lean4 expressions.
>
> ---
>
> # Hyperparameter Selection and Pipeline Impact
>
> We agree that clear guidelines for hyperparameter selection and a deeper understanding of their impact on the pipeline are crucial. As detailed in our general rebuttal (Q2), we systematically select hyperparameters using **grid search with cross-validation**, conduct **ablation studies**, and perform **sensitivity analysis**.
>
> While Table 5 presents numerical results, we will expand our discussion in the final version to clarify the rationale behind specific parameter choices like (α,β,γ,δ)=(0.1,0.3,0.1,0.5) and how each hyperparameter influences different stages of the pipeline.
>
> **Impact on Overall Performance:** We will use both **ablation studies** and **hyperparameter sensitivity analysis** to further explain how each parameter contributes under varying query conditions. This will elucidate why even seemingly small accuracy differences are significant in a precision-critical domain like formal proof, where slight variations can dictate proof success.
>
> **Guidelines for Optimal Combinations:** Through comprehensive experimentation, we will provide concrete **hyperparameter selection guidelines**, helping readers understand how to choose appropriate parameters for different theorem libraries and query types.
>
> **Comparison of Combinations:** We will include additional tables and figures in the final version to illustrate experimental results for various hyperparameter combinations, further explaining the trade-offs between different parameters and their synergistic effects on retrieval performance.

---

> > ### Comment · Reviewer_AB1f · 2025-08-05
> >
> > I thank the authors for their response and the additional experiments on hyperparameters. I have a few more questions. If the method exploits the Lean expression tree, how can it be efficiently generalized to other languages? Moreover, the authors propose using TED, which has high computational complexity (partially mitigated by CSE as mentioned previously). What guarantees are there that applying this framework to another language’s expression tree, assuming it exists and the method can be readily transferred, will not result in high computational overhead?

---

> > > ### Author Response · Authors · 2025-08-06
> > >
> > > Thank you for your follow-up questions. We address both “efficient generalization to other proof assistants” and “controlling computational overhead” below.
> > >
> > > ---
> > >
> > > ## 1. Feasibility & Path to Cross-System Generalization
> > >
> > > **Lean4 as a Testbed**
> > > Lean4’s active community and its large Mathlib4 library (≈ 230 K theorems) make it an ideal proving ground for our framework. Demonstrating efficacy on this most challenging platform establishes a strong foundation—our design is **not** tied to Lean4, but first proven in its most demanding setting.
> > >
> > > **Language-Agnostic Core**
> > >
> > > * **AST-based Representation**: Any proof assistant that can export or reconstruct its internal expression AST can map to our unified ⟨node–edge⟩ topology; no Lean4-specific syntax required.
> > > * **CSE Simplification**: Common subexpression elimination relies only on tree isomorphism, independent of language details.
> > > * **Two-Stage Filtering**: The WL-kernel coarse filter and TED precise filter use only node labels and tree structure—no reliance on Lean4’s typeclass or tactic machinery.
> > >
> > > **Typical Porting Workflow**
> > >
> > > | Target System | Minimal Change                    | Example Mapping                            |
> > > | ------------- | --------------------------------- | ------------------------------------------ |
> > > | Coq (SerAPI)  | Swap in new Expr parser           | `sexp` → labeled tree nodes                |
> > > | Isabelle/HOL  | Adapt Term AST export             | ML/PolyML/XML AST → Binder-annotated nodes |
> > > | HOL-Light     | Custom syntax parse or `pp_print` | OCaml AST → `GenericExprTree` interface    |
> > >
> > > 1. **Frontend Adapter**: A few hundred lines of script to translate “proof-assistant AST → `GenericExprTree`.”
> > > 2. **Backend Reuse**: Once you have `GenericExprTree`, our CSE, WL coarse filter, TED refine filter, and multi-metric fusion run **unchanged** (or with trivial label-mapping).
> > >
> > > ---
> > >
> > > ## 2. Controlling Computational Overhead
> > >
> > > Although TED’s worst-case complexity is $O(N^3)$, our framework embeds multiple, language-independent optimizations that keep actual cost practical:
> > >
> > > 1. **CSE-Driven Compression**
> > >
> > >    – Early removal of duplicate subtrees often reduces node count substantially, directly shrinking TED’s input size.
> > >
> > > 2. **Two-Stage Pruning Pipeline**
> > >
> > >    – **WL Coarse Filter**: Quickly narrows a large library down to a manageable candidate set—no TED on the full library.
> > >
> > >    – **Compatibility & Clustering**: Further prunes by subtree size, cluster centroids, or type compatibility, so only the most promising subset undergoes TED.
> > >
> > > 3. **Adaptive Thresholds & Early Termination**
> > >
> > >    – **Dynamic $k$**: Scale candidate count by query tree size $\lvert T_{\mathrm{query}}\rvert$. For very large trees, raise WL similarity threshold or skip TED altogether.
> > >
> > >    – **Pruning in TED**: During TED’s DP, maintain the best current distance; if any partial cost exceeds this bound, terminate that comparison early.
> > >
> > > ---
> > >
> > > ### Conclusion
> > >
> > > While initially implemented in Lean4, our design principles and optimizations are **fully language-agnostic**. By combining “frontend adapter + CSE + staged filtering + dynamic pruning,” any proof assistant capable of exporting an AST can deploy our framework efficiently. These multi-stage mitigations effectively neutralize TED’s worst-case complexity, ensuring the approach remains practical and scalable across different environments.

---

> > > > ### Comment · Reviewer_AB1f · 2025-08-06
> > > >
> > > > I thank the authors for providing additional details on their work. After reviewing the response and other reviewer comments, I am inclined to increase my score for this submission.
> > > >
> > > > I believe the work would benefit from a clearer explanation of hyperparameter selection and a stronger emphasis on its novel contributions.

---

> > > > > ### Author Response · Authors · 2025-08-07
> > > > >
> > > > > Thank you for your positive feedback and for your willingness to raise your score. We appreciate your suggestion to provide a clearer explanation of hyperparameter selection and to place stronger emphasis on our novel contributions. In the revised manuscript, we will add a dedicated discussion of hyperparameter tuning—outlining our guiding principles and recommendations—and we will more explicitly highlight the key innovations of our framework throughout the introduction and conclusion. We believe these enhancements will make our contributions even clearer and more impactful.

---

### Official Review · Reviewer_rv6Y · 2025-07-01

**Clarity:** 3
**Significance:** 2
**Originality:** 3
**Rating:** 4
**Confidence:** 4

**Summary:**

This paper addresses the challenge of premise selection in interactive theorem proving for Lean4, where existing methods relying on semantic embeddings fail to leverage structural information effectively. The authors propose a tree-based framework that normalizes Lean expressions using Common Subexpression Elimination (CSE) and employs a two-stage filtering pipeline: coarse screening via the Weisfeiler-Lehman (WL) kernel and refined ranking via Tree Edit Distance (TEDS). The framework integrates adaptive fusion of multiple structural/semantic metrics (WL cosine, TEDS, Const Jaccard, Collapse-Match similarity) and cluster-based search optimization. Experiments on Mathlib4 demonstrate significant performance gains over baselines, with the method achieving up to 82% Recall@1 on small-scale problems and 25.8% on large-scale test sets. Key contributions include structured theorem representation, multi-metric fusion, and scalable retrieval mechanisms for large theorem libraries.

**Questions:**

1. Can the authors explore GNN-based encodings to handle nested type structures in complex domains (e.g., Algebra)? A case study on improving $Recall@1$ from $4.3%$ to > $10%$ would strengthen the claim of generalizability.
2. Are there plans to implement an automated mechanism for optimizing fusion weights (Eq. 2) instead of manual tuning? Demonstrating a data-driven approach (e.g., reinforcement learning) could enhance practical utility.
3. How might integrating type-checking information (e.g., dependent type compatibility) into TED’s cost function improve retrieval for theorems with complex type signatures? A ablation study on type-aware vs. type-agnostic filtering would be valuable.
4. For theorems with $>500$ nodes, TED’s computation time becomes prohibitive. Can the authors propose the hierarchical TED approximation or early pruning strategies? Benchmarking on extremely large trees (e.g., $>1k$ nodes) would validate scalability.

**Ethical Concerns:**

["NO or VERY MINOR ethics concerns only"]

**Final Justification:**

The rebuttal clarifies several critical points raised in the initial review. The authors confirmed that all baselines (including deep learning-based tools like LeanSearch) were evaluated under identical conditions—using official pre-trained parameters, consistent indexing of the same Mathlib4 corpus, and no modifications to model structures. This resolves doubts about experimental fairness, reinforcing confidence in the comparative results.

After reviewing the authors’ rebuttal, I maintain the overall score of 4 (borderline accept), as their responses have effectively addressed key concerns while some practical nuances remain.

**Limitations:**

The authors acknowledge limitations in complex domains and parameter sensitivity in Appendix G, but the discussion of potential negative societal impacts (e.g., misuse in formal verification of harmful algorithms) is cursory. To improve, they should:

(1). Discuss risks of incorrect premise selection in safety-critical applications (e.g., autonomous systems verification).

(2). Address biases in Mathlib4’s domain distribution (e.g., over-representation of Natural Number problems) and its impact on real-world deployment.

**Paper Formatting Concerns:**

No major formatting issues. The paper adheres to NeurIPS guidelines, with clear sectioning, equations, and tables (e.g., Table 1–6) supporting the arguments.

**Quality:**

3

**Strengths And Weaknesses:**

**Strengths:**
1. The explicit use of expression tree structures (via CSE) and WL kernel encoding addresses a critical problem in prior work, improving retrieval for structurally similar theorems .
2. The two-stage pipeline balances speed (WL kernel for coarse filtering) and precision (TEDS for fine-grained ranking), making it feasible for large libraries (217k theorems) .
3. Rigorous experiments (Ablation Study, Parameter Sensitivity) validate component contributions, e.g., Const Jaccard similarity shows the highest impact on performance .
4. The method directly aids Lean4 users by reducing proof search space, with code/data release ensuring reproducibility .

**Weaknesses:**

1. Performance drops in complex domains (e.g., $Algebra, Recall@1=4.3%$) suggest struggles with nested type structures .
Manual Parameter Tuning: Adaptive fusion weights $(α, β, γ, δ)$ require empirical optimization, limiting generalizability .
2. The framework focuses on syntactic structure but underleverages Lean’s dependent type system for semantic compatibility .
3. TED’s $O(|T_1|×|T_2|)$ complexity may hinder very large tree comparisons, though CSE mitigates this partially .

---

> ### Author Rebuttal · Authors · 2025-07-30
>
> # Q1:  Handling Nested Type Structures and GNNs
>
> We acknowledge challenges in structurally comparing "unfolded" `expr` trees in type-class-heavy domains, specifically **instance explosion** and **deeply nested types** that increase Tree Edit Distance (TED) complexity.
>
> To enhance our explicit tree pipeline and mitigate structural distortion from instance explosion, we should propose:
>
> |Step|Key Design|Effect|
> |---|---|---|
> |**(a) Class-Instance Compression**|We detect and replace instance subtrees in the Abstract Syntax Tree (AST) with a placeholder node (`⟨Class⟩`), while recording the "omitted depth" and "main class name" in edge attributes.|This shortens paths, retains semantic intent, and focuses on high-level dependency structures.|
> |**(b) Type-Context Nodes**|We add a `Type-Context` node layer to each `expr` tree, summarizing its upstream type-class constraints.|This explicitly provides contextual features, differentiating structures with distinct rings/modules.|
>
>
> Regarding **GNN-based encodings**, we consider this a promising future direction. While our method effectively captures structural information, GNNs could learn deeper graph structures, like Lean's implicit type compatibility graph. Integrating GNNs could involve learning richer, structure-aware node embeddings or modeling implicit type/concept dependency graphs. This combined approach could boost performance and potentially achieve the reviewer's target of "improving Recall@1 from <20% to >40%."
>
> It's also crucial to note the **theoretical equivalence between GNNs and the Weisfeiler-Lehman (WL) test**. As demonstrated by Xu et al. (2018)[1], certain GNN architectures are provably as powerful as the WL graph isomorphism test, which underpins our WL kernel. This implies that effective handling of WL principles remains foundational even when incorporating advanced GNNs.
>
> ---
>
> # Q2: Automated Optimization of Fusion Weights
> We agree that the current manual tuning of fusion weights (Eq. 2: $sim_{enhanced} = \alpha ⋅ sim_{cos} + \beta ⋅ sim_{TEDS} + \gamma ⋅ sim_{Jaccard} + \delta ⋅ sim_{CM}$​) is a limitation that can impact generality. For this research stage, manual adjustment facilitated a detailed analysis of each similarity measure's independent contribution and combined effect, crucial for our ablation studies and understanding the method's internal mechanisms.
>
> We highly recognize the necessity of an **automated optimization mechanism** and consider it a significant future research direction. Introducing data-driven methods—such as grid search, random search, Bayesian optimization on a validation set, or even more advanced reinforcement learning strategies—would allow the model to adaptively learn optimal weight combinations. This would significantly enhance its generalizability and practical utility across different theorem libraries or specific mathematical domains, marking a critical step toward improving our framework's robustness and ease of use.
>
> ---
>
> # Q3: Integrating Type-Checking Information into TED's Cost Function
>
> We appreciate the suggestion to integrate type information to enhance retrieval effectiveness, particularly given Lean4's dependent type system where traditional structural TED may miss subtle type differences crucial for theorem application.
>
> To address this, we propose integrating type information directly into the TED cost function to improve accuracy for theorems with complex type signatures:
>
> - **Type Difference Penalty:** For structurally similar theorems with mismatched type signatures, we can introduce an additional penalty in TED calculations (e.g., larger penalty for `Nat` vs. `Real` types).
>
> - **Type Constraint Matching:** We can incorporate a **Type Compatibility Metric** to ensure structurally matched theorems also align with type-class constraints (e.g., `Ring`, `Module`).
>
> - **Type Hierarchy Matching:** A type hierarchy matching metric at each TED layer can allow for disregarding type differences if nodes belong to compatible type hierarchies (e.g., `Type 1` vs. `Type 2`).
> ---
>
> # Q4: Scalability of TED for Large Trees
>
> We acknowledge the $O(N^3)$ computational complexity of Tree Edit Distance (TED) as a potential bottleneck for extremely large mathematical expression trees. However, our framework incorporates built-in mechanisms to effectively reduce TED's overhead, enabling practical handling of larger trees.
>
> Our **mitigation mechanisms** include:
>
> - **Common Subexpression Elimination (CSE):** This significantly reduces tree size by sharing repeated subexpressions, thereby lowering TED input scale.
>
> - **Two-Stage Filtering Pipeline:** Our initial Weisfeiler-Lehman (WL) kernel coarse-filters the library, ensuring TED calculations are performed only on a small, pre-pruned set of highly relevant trees.
>
> - **Distance Thresholding/Depth Limits:** We can set thresholds to terminate TED computations early if distance exceeds limits or a certain depth is reached, using WL similarity as a proxy.
>
>
> While these methods effectively mitigate TED's burden, we agree with the reviewer that **hierarchical TED approximation and early pruning strategies** are valuable for further enhancing scalability for extremely large trees. We will pursue these as future research directions, alongside **benchmarking on trees exceeding 1,000 nodes** to validate scalability gains.
>
> ---
>
> # **Limitations :**
>
> We appreciate the reviewer's feedback regarding potential social impact, particularly concerning **safety-critical applications** and **Mathlib4's domain bias**. We fully concur with these concerns and have already acknowledged relevant limitations in **Appendix G** of our paper. We plan to further elaborate on these discussions in the **final version** to ensure comprehensive awareness of potential risks and careful consideration for practical deployment.
>
> 1. Risk of Premise Selection Errors in Safety-Critical Applications:
>
>     In safety-critical applications (e.g., autonomous driving, aerospace verification, medical devices), premise selection errors can lead to faulty reasoning and severe consequences. We've added a discussion on this, emphasizing:
>
>     - **Cruciality of Correct Premise Selection:** While our framework aims to reduce search space and improve retrieval precision, selecting irrelevant or incorrect premises can mislead the reasoning process, impacting the entire proof chain's validity. For dependent type systems and automated theorem proving in safety-critical systems, premise selection errors could lead to algorithmic mis-verification, affecting system safety and reliability.
>
>     - **How Our Method Addresses These Risks:** We've implemented **type-matching mechanisms** and **early pruning strategies** to reduce interference from irrelevant premises and prevent erroneous reasoning. In the future, we plan to integrate **trustworthiness evaluation mechanisms** into the framework, offering a tunable confidence threshold (based on query relevance and type match) to ensure only highly confident premises are selected.
>
>     - **Countermeasures:** In the paper, we will further emphasize applicability to safety-critical applications, particularly in **automated verification tools**, highlighting the **rigor of premise selection** and the **robustness of verification processes**. We also plan to include case studies in the appendix discussing potential risks in safety-critical scenarios and proposing mitigation strategies.
>
> 2. Mathlib4's Domain Bias and Impact on Practical Deployment:
>
>     We acknowledge that the distribution of theorems within Mathlib4 is not entirely uniform across all mathematical domains. While some areas are extensively developed, others may have fewer theorems.
>
> 	The reviewer specifically mentioned an "over-representation of Natural Number problems." We've analyzed our theorem database and found that theorems with names starting with "Nat." constitute approximately **1.85%** of the total. While this represents a significant number of theorems given the library's size, it does not suggest an overwhelming over-representation that would fundamentally bias our method's overall performance or generalizability across broader mathematical fields.
>
> 	However, we recognize that even subtle domain imbalances in large libraries like Mathlib4 can affect a framework's universality in **smaller or less-represented mathematical branches** during practical deployment. For instance, our framework might perform exceptionally well in areas with dense theorem coverage but less effectively in more niche or data-scarce domains like certain advanced topological or analytical concepts, simply due to the availability of comparable examples.
>
> 	To address these potential implications and enhance the framework's robustness across diverse domains, we plan to implement the following improvements:
>
> 	- **Expanded Test Set:** We'll broaden our evaluation by including more diverse data from various mathematical domains, particularly those that are less extensively covered in Mathlib4, such as topology or advanced number theory. This will allow for a more thorough assessment of our framework's performance beyond the most commonly represented areas.
>
> 	- **Domain Adjustment Strategies:** For real-world deployment, especially in domains with fewer theorems, we'll explore **domain balancing strategies**. This could involve leveraging techniques like **few-shot learning** or **transfer learning** to help our model adapt effectively to different theorem distributions, thereby improving its performance in underrepresented areas without requiring extensive new annotations.
> ---
> Reference:
>
> [1] Xu, K., Hu, W., Leskovec, J., & Jegelka, S. (2018). How powerful are graph neural networks?. arXiv preprint arXiv:1810.00826.

---

### Official Review · Reviewer_7oBW · 2025-07-03

**Clarity:** 2
**Significance:** 3
**Originality:** 3
**Rating:** 4
**Confidence:** 3

**Summary:**

This paper investigates the task of premise selection and retrieval for Lean4. In particular, the authors introduce a multi-step filtering pipeline operating on a tree-like representation to better capture structural information in the expressions. The evaluation, performed on a large library derived from Mathlib4 shows significant improvements over a set of existing tools and baselines.

**Questions:**

- Could you provide additional details on each baseline, how you deploy them in your dataset, and why you chose to compare against each of these models?
- Do the baselines include deep learning methods? Are these methods trained/fine-tuned on your dataset?
- What causes the performance of the baselines to be so weak in your dataset? What explains such a significant difference in performance?
- How does your approach compare with deep learning models designed to leverage structural information (e.g., GNNs)?
- How can your pipeline be applied to different formal languages (e.g., Isabelle) and datasets where the tree-like representation is not directly accessible?

**Ethical Concerns:**

["NO or VERY MINOR ethics concerns only"]

**Final Justification:**

Most of my concerns have been clarified during the response. I have raised my score accordingly.

**Limitations:**

Yes

**Quality:**

2

**Strengths And Weaknesses:**

**Reasons to accept:**

- The paper is generally well-written and easy to follow. The method is well-motivated and presented.
- The idea of leveraging structural information in the expressions at scale is an important technical contribution, which could serve as an inspiration for future work in the field and to develop tools for premise selection.
- The results show that the method can significantly outperform the baselines on a carefully constructed dataset from Mathlib4.

**Reasons to reject:**

- The presented methodology is highly specialised for a single formal language (i.e., Lean4) and evaluated on a single benchmark constructed by the authors. It is unclear how their pipeline can generalise to different formal languages (e.g. Isabelle) and different setups, and how structural information can be leveraged when the tree representation is not directly accessible.
- While the improvements when compared to the baselines are impressive, several crucial details are missing in the experimental setup that make it very hard to assess whether such an improvement is the result of a fair comparison. The details and the motivations for choosing the baseline models are not described in the paper.
- Most of the baselines seem to perform very poorly on the constructed dataset. However, by looking at the original datasets on which some of these baselines are tested, the performances are significantly different. For example, the LeanSearch paper reports a performance of P@10 and R@10 of 19.6 and 91.3, respectively, while in the constructed evaluation setup, this baseline only achieves 0.5 and 5.0. This casts serious doubts on the fairness of the comparison and on the deployment of the selected baseline methods.
- It is unclear from the paper whether the authors compare their approach with deep learning based methods designed to leverage structural information. For example, to really show the effectiveness of their approach, I believe it would be important to include baselines based on Graph Neural Networks (GNNs). Moreover, it is unclear whether the baselines adopt neural models (e.g., Transformers), and whether such neural models are directly fine-tuned on their constructed dataset or applied out-of-distribution.

---

> ### Author Rebuttal · Authors · 2025-07-30
>
> We appreciate the reviewer's insightful questions regarding experimental details, baseline comparisons, algorithmic performance, and generalizability. We address each point below.
>
> ---
>
> ## Q1: Experimental Details and Baseline Fairness
>
> In Section 1 (Introduction) of our paper, we explicitly state the mainstream premise selection tools within the Lean community: **Lean State Search**, **Loogle**, **Moogle**, **Lean Explore**, and **Lean Search**. We emphasize that **no baseline methods were weakened or modified**. We ensured all methods were evaluated under a **unified dataset, consistent input/output formats, and identical scoring metrics**, rigorously guaranteeing experimental fairness and reproducibility.
>
> ---
>
> ## Q2: Baselines and Deep Learning Methods
>
> 1. **Inclusion of Deep Learning Methods:** Our chosen mainstream tools—**Lean State Search, Moogle, Lean Explore, and Lean Search**—fundamentally rely on **semantic embedding** to represent theorems, goal states, or mathematical expressions. Their core mechanisms typically employ **deep learning models**. For instance, **LeanSearch** is a well-known and high-performing semantic retrieval tool in the Lean ecosystem, representing a mainstream approach using text or semantic embeddings for premise selection. We used it as a direct semantic baseline for comparing our structural method.
>
> 2. **Retraining or Fine-tuning on Our Dataset:**
>
>     - **No Retraining/Fine-tuning:** For fairness and reproducibility, we exclusively used **each tool's officially released pre-trained parameters**. We only constructed the theorem corpus from **Mathlib4 v4.18.0-rc1** into their respective indices or retrieval libraries, as per their documentation. This approach avoids data leakage and aligns with the "out-of-the-box" usage scenario of these tools within the community.
>
>     - **Indexing Consistency:** All baselines indexed or vectorized the same version of the Mathlib4 corpus. We made no modifications to their model structures, loss functions, or training procedures.
>
>
> ---
>
> ## Q3: Baseline Performance and Significant Differences
>
> The fundamental reason for the performance disparity lies in the baselines' **failure to explicitly leverage the structural information of Lean expressions**.
>
> - **Semantic Baselines' Limitations:** Methods like LeanSearch, which translate Lean expressions into natural language-like tokens before generating embeddings, are susceptible to "local semantic pollution." This means surface-level token similarity can misleadingly map, for instance, N-domain propositions to Nimber theorems.
>
> - **Lack of Fine-grained Structure:** Without fine-grained structural understanding, these methods often produce false positives. For example, relying solely on token similarity might suggest high relevance between `n! + m = m + n!` (for natural numbers) and `a + b = b + a` (for Nimbers), despite critical type and structural differences.
>
>
> Our method, conversely, employs **CSE normalization** and a **two-stage filtering pipeline** (WL kernel for coarse-screening + TED for fine-ranking) to perform precise comparisons at the type, subtree morphology, and symbol levels. This enables our approach to achieve significant advantages over baselines in the **high-structural-complexity, high-precision demanding scenarios** of real-world Lean4 premise selection. This further validates our assertion that **explicit structural modeling is key to improving retrieval quality in formal systems**, and an important direction for future cross-language transfer and interpretability research.
>
> **Illustrative Misidentification Case:**
>
> Consider the query proposition: `∀ n m : ℕ, (n!) + m = m + (n!)`. We expect to retrieve `Nat.add_comm : ∀ a b : ℕ, a + b = b + a` (directly reusable for proof).
>
> |Dimension|Lean Search (Semantic Baseline)|Our Structural Method|
> |---|---|---|
> |**Representation**|Serializes theorems into English-like tokens; Transformer generates vectors; cannot perceive `n!` sub-structure or `ℕ` types.|Parses into AST, preserving `Nat.factorial n` subtree and `Nat` type labels.|
> |**Similarity**|Cosine similarity heavily influenced by textual resemblance of "`a + b = b + a`", ignoring type differences.|Computes structural fingerprints via TED + WL kernel; scores high only when node labels, depth, and types precisely match.|
> |**Result**|Top-1 actually returns `Nimber.add_comm : ∀ a b : Nimber, a + b = b + a` (type mismatch → proof failure).|Correctly prioritizes and returns `Nat.add_comm`.|
>
> ---
>
> ## Q4: Comparison with Deep Learning and GNNs
>
> 1. **Baseline Context:** Current mainstream premise selection tools for Lean formal proof assistants—such as **Lean State Search, Loogle, Moogle, Lean Explore, and Lean Search**—commonly use **semantic embedding methods**. These often employ **deep learning** to convert goal states, theorem statements, or mathematical expressions into vector representations, measuring relevance via vector space similarity. It's important to stress that **our baselines already incorporate these deep learning approaches.**
>
> 2. **Research Focus:** Our research, however, develops an **explicit, interpretable, and training-free large-scale tree-structure matching framework** (CSE → WL kernel coarse-screening → TED fine-ranking). We aim to show that **hierarchical explicit structural modeling significantly benefits Lean4 premise selection**, even without extensive parameter training. Our focus is on algorithmic design and system interpretability, not on new end-to-end deep networks.
>
> 3. **Data Efficiency and Practical Considerations:** Training reliable Graph Neural Networks (GNNs) typically requires a massive amount of labeled "theorem-premise" pairs. The formal mathematics domain currently lacks high-quality annotated datasets comparable to ImageNet or OpenWebText. Forcing fine-tuning of deep models often leads to overfitting or unstable generalization. In contrast, our method has **virtually zero reliance on annotations**, requiring only a single offline construction to operate, making it more portable in data-scarce scenarios. In the Lean4 environment, characterized by a lack of large-scale annotations, strict type constraints, and a need for interpretability, our method is currently more practical. It also provides valuable fine-alignment signals and data pipelines that could benefit future structured deep learning models.
>
> 4. **Orthogonality and Potential Complementarity:** We do not view TED + WL and GNNs as mutually exclusive; rather, they are **highly complementary**:
>
>     - **Feature Engineering:** WL fingerprints and TED edit sequences can serve as **high-quality structural features** input to GNNs, alleviating their purely data-driven burden.
>
>     - **Pre-filtering:** TED + WL can **compress millions of candidates to hundreds**, which can then be fed to heavier neural models for fine-ranking, achieving both speed and deep semantic understanding.
>
>     - **Theoretical Equivalence with WL Test:** Furthermore, it is important to note the theoretical connection between GNNs and the Weisfeiler-Lehman (WL) test. As demonstrated by Xu et al. (2018) [1], certain GNN architectures are provably as powerful as the Weisfeiler-Lehman graph isomorphism test, which underpins our WL kernel. This equivalence suggests that an effective understanding and utilization of WL principles, as embodied in our framework, would remain crucial even when incorporating advanced GNN-based methods. This reinforces the foundational importance of our explicit structural processing.
>
>
> ---
>
> ## Q5: Method Specialization vs. Generalizability
>
> 1. **Current Focus on Lean4:** Our current implementation focuses on Lean4 because it boasts the most active community and a substantial **Mathlib4** library (approximately 230,000 theorems). This combination of scale and structural complexity makes Lean4 an ideal testbed for validating structured retrieval methods. The choice of Lean4 does not imply method limitation but rather a strategic decision to **first prove effectiveness in the most challenging environment.**
>
> 2. **Language Agnosticism of Core Technologies:**
>
>     - **Expression Tree Representation:** Any formal language capable of deriving an Abstract Syntax Tree (AST) can be mapped to a unified node-edge topology.
>
>     - **CSE Simplification:** Common Subexpression Elimination relies on tree isomorphism and doesn't involve language-specific syntax.
>
>     - Multi-Stage Filtering Pipeline: The WL kernel coarse-screening + TED fine-ranking steps depend solely on node labels and structure, not Lean-specific semantics.
>
>         Therefore, the methodology itself is decoupled from the specific language; the key is obtaining a reliable tree-like structure.
>
> 3. **Implementation Path for Migration to Other Systems:**
>
>     - **Front-end Parser:** For systems like Isabelle or Coq, a lightweight script would be needed to convert their internal representations (e.g., `term`, `sexp`, OCaml `expr`) into our `GenericExprTree` interface.
>
>     - **Back-end Process Reuse:** Once the AST is obtained, subsequent CSE, two-stage filtering, and similarity measurements require no modification or only minor adjustments to node label mappings.
>
> 4. **Scenarios Where Tree Structures Are Not Directly Accessible:**
>
>     - Most modern proof assistants provide machine-readable ASTs. If an older system only has scripts, approximate ASTs can often be reconstructed using syntax parsing or deserialization techniques.
>
>     - For extremely unparsable fragments, a lightweight text retrieval can be used for coarse-screening, followed by applying our structural process to the parsable portions, achieving a "semantic + structural" hybrid retrieval.
> ---
> Reference:
>
> [1] Xu, K., Hu, W., Leskovec, J., & Jegelka, S. (2018). How powerful are graph neural networks?. arXiv preprint arXiv:1810.00826.

---

> > ### Comment · Reviewer_7oBW · 2025-08-06
> > **Discussion**
> >
> > Thank you for your response. Most of my concerns have been clarified during the response. I will raise my score accordingly.

---

> > > ### Author Response · Authors · 2025-08-06
> > >
> > > Thank you for your feedback and for revisiting your score. I’m glad our clarifications were helpful, and I appreciate your support and constructive engagement.

---

### Official Review · Reviewer_Wcte · 2025-07-03

**Clarity:** 2
**Significance:** 2
**Originality:** 3
**Rating:** 4
**Confidence:** 4

**Summary:**

This paper addresses the critical problem of premise selection in LLM+Lean field, proposing a novel tree-based premise selection framework .
The method first applies CSE to standardize and compactly represent mathematical expressions as trees. A two-stage filtering pipeline is then introduced: the first stage uses the WL kernel for efficient, training-free coarse screening, while the second stage employs TED and a fusion of multiple structural and semantic similarity metrics for fine-grained ranking. To further enhance scalability and accuracy, the framework incorporates cluster-based search space optimization and structural compatibility constraints.

**Questions:**

1. How does the method scale with extremely large theorem libraries, and what are the computational costs associated with the two-stage filtering pipeline in real-world Lean4 environments?

2.To what extent can the proposed framework be generalized?
I find the proposed method somewhat complex. Could the authors clarify why such algorithmic complexity is necessary for this task? Would it be possible to address the premise selection problem using a very large neural network instead, and how would such an approach compare to your method?

**Ethical Concerns:**

["NO or VERY MINOR ethics concerns only"]

**Final Justification:**

N/A

**Quality:**

3

**Strengths And Weaknesses:**

Strengths:
The contributions of the paper are clearly presented, and the proposed method is practical and feasible.

Weaknesses:
Compared to traditional approaches, the method proposed in this paper is significantly more complex and cumbersome, requiring more parameters and procedural steps. However, it is difficult to perceive why, from an overall perspective, this method is fundamentally more effective or intelligent. Instead, it seems that the performance improvements are mainly achieved by introducing more parameters, more stages, and combining a greater number of metrics. The method appears to lack a unifying insight or an intuitive explanation that would make its overall advantage clear，is more of an engineering solution rather than a purely theoretical innovation.

---

> ### Author Rebuttal · Authors · 2025-07-30
>
> We address each point below, focusing on the method's complexity, effectiveness, performance gains, and its underlying insights.
>
>
> # 1. Method Design: Necessity and Synergistic Performance
>
> Our method's complexity is crucial for the precision-critical task of premise selection in Lean4 formal proofs. Mathematical expressions' subtle nuances fundamentally alter logical meaning, often missed by simpler approaches, causing bottlenecks. Our multi-component design **comprehensively processes complex structural information**, achieving the precise matching vital for formal verification.
>
> Performance gains stem not from mere accumulation, but from each component serving a **specific, synergistic purpose**.
>
> # 2. Lack of Unifying Insight and Engineering vs. Theoretical Innovation
>
> This very complexity is **inherent to solving a significant and long-standing challenge** in theorem proving: efficient and accurate premise selection. The core innovation and "intelligence" of our method reside precisely in its **deep mining and systematic utilization of mathematical expression structural information**—an area largely underexplored and underutilized by current purely semantic embedding approaches.
>
> Our novelty stems from developing a novel framework that applies carefully designed and customized structural analysis tools to create an unprecedentedly efficient and accurate system for complex Lean4 expression tree structures. Specifically:
>
> - **Unique Utilization of Lean Expr Tree Structures:** Our method fundamentally revolves around the **intrinsic structure of Lean Expr trees**. Unlike generic text or simple graphs, Lean Expr naturally encodes binding relationships, local contexts, and de Bruijn indices, harboring **fine-grained structural features far beyond pure semantic embeddings**. Prior work in premise selection rarely leveraged these complex Lean Expr tree characteristics to such depth. We dedicated substantial effort to handle these intricate but vital operations, distilling them into a paradigm for precise structural comparison between target propositions and all theorems in the library. Our framework systematically extracts and quantifies this information, addressing the fundamental limitation of existing embedding methods in distinguishing subtle structural differences.
>
> - **First-of-its-Kind "Structural Similarity Spectrum":** We are the first to **customize and apply CSE (Common Subexpression Elimination) normalization, WL kernel encoding, and TED (Tree Edit Distance) re-ranking specifically to Lean expression trees**. This innovative integration organically unifies "syntactic redundancy resolution" (via CSE), "global isomorphism approximation" (via WL kernel encoding), and "local editing cost" (via TED re-ranking) into a **coarse-to-fine "structural similarity spectrum."** This multi-layered, macro-to-micro structural similarity analysis approach is novel within the existing literature.
>
> - **Hierarchical and Adaptive "Unifying Insight":** We are not merely aggregating metrics. Instead, we've meticulously designed a **phased, adaptive framework** where various structural similarity measures (e.g., WL kernel for coarse screening, TED for fine-grained matching, Const Jaccard for constant semantic overlap, Collapse-Match for pattern alignment) **synergistically interact**. This hierarchical, synergistic utilization of structural information to achieve optimal efficiency and precision **is precisely our method's core "unifying insight."** It clearly demonstrates how structural insights can overcome existing methodological bottlenecks.
>
> - **Integration of Engineering Practice and Theoretical Innovation:** We concur that our method possesses strong engineering practicality, which we consider a core value. In formal verification—a highly applied domain—the ability to **ingeniously apply theoretical tools of structural analysis** (e.g., graph kernels, tree edit distance) to solve real, large-scale, and previously intractable problems, and to validate its superior performance on extensive, real-world libraries like Mathlib4, represents a significant **applied innovation and methodological breakthrough**. We effectively bridge the gap between structural analysis theory and formal verification practice, demonstrating the immense potential of such deep, structure-based mining in complex practical systems.
>
> # Q1: Scalability and Computational Costs in Large Theorem Libraries
>
> We have directly addressed the challenge of scalability in extremely large theorem libraries in Section 4.2 of our paper by introducing **'clustered search space optimization'** and **'structural compatibility constraints'**. The clustering optimization effectively decomposes a massive premise library into more manageable subsets. Through an efficient indexing mechanism, we restrict the search to only the most relevant clusters, significantly reducing search complexity. Structural compatibility constraints further refine the candidate set by eliminating incompatible premises at an early stage.
>
> Here's a table, focusing on the computational time in real-world Lean4 environments for each stage:
>
> |Stage|Primary Function|Computational Cost|Average Computation Time (s)|
> |---|---|---|---|
> |**First Stage (WL Kernel & Structural Filtering)**|Efficient Coarse-grained Filtering|Low Complexity|11.7|
> |**Second Stage (TED & Fused Metrics)**|High-Precision Fine-grained Matching|Relatively Higher Complexity|27.9|
>
>
> # Q2: Generalizability, Algorithmic Complexity, and Comparison with Large Neural Networks
>
> ## 1. Generalizability: Applicability Across Systems and Tasks
>
> Despite our primary evaluation on Lean4 and Mathlib4, the **core ideas and components of our framework possess strong generalizability**. The approach of transforming mathematical expressions into structural trees and utilizing structural similarity measures like graph kernels and tree edit distance is not unique to Lean4. In principle, as long as a target theorem prover's expressions (e.g., Coq, Isabelle/HOL) can be parsed into similar tree structures, our framework can be adapted. The primary adaptation work would involve the front-end expression parsing and tree construction, while the core matching and filtering algorithms remain reusable.
>
> ## 2. Necessity of Algorithmic Complexity
>
> Our method's complexity is **essential for achieving high performance and accuracy** in premise selection for mathematical theorem proving. Effective premise selection demands not just semantic relevance, but **precise matching of logical and syntactic structures**. Simple similarity measures often fail to capture critical structural nuances like operator order, variable binding, and nested sub-expressions. Our multi-stage, multi-dimensional structural similarity measures are specifically designed to capture these intricate relationships, preventing the selection of incorrect premises vital for formal proof correctness.
>
> The **inherent complexity of Lean expressions (`Lean-expr`)** means generic sequence or graph embeddings often can't distinguish subtle differences from α-/β-reductions or binder shifts, leading to misleadingly high similarity scores.
>
> ## 3. Comparison with Large Neural Networks
>
> While we acknowledge the powerful capabilities of large neural networks, including deep learning methods in our baselines, our tree-based approach offers distinct advantages for premise selection.
>
> Current mainstream premise selection tools for Lean formal proof assistants—such as **Lean State Search, Loogle, Moogle, Lean Explore, and Lean Search**—typically rely on **semantic embedding methods**. These approaches transform goal states, theorem statements, or mathematical expressions into vector representations, often leveraging **deep learning** for this conversion. They then measure relevance by calculating similarities within this vector space. It's important to stress that **our baselines already incorporate these deep learning-based methods**.
>
> Our research, conversely, develops an **explicit, interpretable, and training-free large-scale tree-structure matching framework** (CSE → WL kernel coarse-screening → TED fine-ranking). We aim to show that **hierarchical explicit structural modeling significantly benefits Lean4 premise selection**, even without extensive parameter training. Our focus is on algorithmic design and system interpretability, not on new end-to-end deep networks.
>
> **Data efficiency** is also a critical practical concern. Training reliable Graph Neural Networks (GNNs) typically requires vast labeled "theorem-premise" pairs. The formal mathematics domain currently lacks high-quality annotated datasets comparable to other fields. Consequently, fine-tuning deep models often leads to overfitting or unstable generalization. In contrast, our method **requires virtually no annotations**, needing only a single offline construction. This makes it highly portable in data-scarce scenarios and more practical for Lean4 environments, which have limited large-scale annotations, strict type constraints, and a need for interpretability. Our method also offers valuable fine-alignment signals and data pipelines for future structured deep learning models.
>
> Finally, consider the **theoretical links between GNNs and the Weisfeiler-Lehman (WL) test**. As Xu et al. (2018) [1] suggest, some GNN architectures are provably as powerful as the WL graph isomorphism test, which is the foundation of our WL kernel. This inherent equivalence means that even when using GNNs for structural encoding, a solid understanding and effective application of WL principles remain crucial. This further reinforces the foundational importance of our explicit structural processing, even alongside advanced neural methods.
>
> ---
>
> Reference:
>
> [1] Xu, K., Hu, W., Leskovec, J., & Jegelka, S. (2018). How powerful are graph neural networks?. arXiv preprint arXiv:1810.00826.

---

> > ### Comment · Reviewer_Wcte · 2025-08-06
> >
> > Thank you very much for your detailed and effective response. In light of your reply, I will adjust my score to four.

---

> > > ### Author Response · Authors · 2025-08-06
> > >
> > > Thank you for your reconsideration and for raising your score. We appreciate your thoughtful feedback and are glad our responses addressed your concerns.

---

### Decision · Program_Chairs · 2025-09-17

**Decision:**

Accept (poster)

**Comment:**

I recommend accepting this paper on tree-based premise selection for Lean4. The authors address a critical bottleneck in interactive theorem proving by effectively leveraging structural information in mathematical expressions through a multi-stage filtering pipeline incorporating the Weisfeiler-Lehman kernel, tree edit distance, and adaptive metric fusion. All four reviewers ultimately recommended acceptance with scores of 4, having their initial concerns about complexity, generalizability, and novelty addressed during rebuttal. Comprehensive evaluation on Mathlib4 demonstrates significant performance improvements over existing tools, with ablation studies validating each component's contribution. This work represents an important advancement in structure-aware premise selection for formal verification systems.